



# Assimilating near real-time mass balance observations into a model ensemble using a particle filter

Johannes M. Landmann[1,2], Hans R. Künsch[3], Matthias Huss[1,2,4], Christophe Ogier[1,2], Markus Kalisch[3], and Daniel Farinotti[1,2]

[1]Laboratory of Hydraulics, Hydrology and Glaciology (VAW), ETH Zurich, Zurich, Switzerland
[2]Swiss Federal Institute for Forest, Snow and Landscape Research (WSL), Birmensdorf, Switzerland
[3]Seminar for Statistics, ETH Zurich, Zurich, Switzerland
[4]Department of Geosciences, University of Fribourg, Fribourg, Switzerland

**Correspondence:** Johannes M. Landmann (landmann@vaw.baug.ethz.ch)

**Abstract.** Glaciers fulfil important short-term functions like water supply for drinking and hydropower, and they are important indicators of climate change. This is why the interest in near real-time mass balance nowcasting is high. Here, we address this interest and provide an evaluation of seven continuous observations of point mass balance based on on-line cameras transmitting images every 20 minutes on three Swiss glaciers during summer 2019. Like this, we read 352 near real-time daily point mass balances in total from the camera images, revealing melt rates of up to $0.12$ meter water equivalent per day ($\mathrm{m\,w.e.\,d^{-1}}$) and the biggest total melt on the tongue of Findelgletscher with more than $5\,\mathrm{m\,w.e.}$ in 81 days. These observations are assimilated into an ensemble of three temperature index (TI) and one simplified energy balance mass balance models using an augmented particle filter with a custom resampling method. The state augmentation allows estimating model parameters over time. The custom resampling ensures that temporarily poorly performing models are kept in the ensemble instead of being removed during the resampling step of the particle filter. We analyse model performance over the observation period, and find that the model probability within the ensemble is highest on average with 58% for an enhanced TI model, a simple TI model reaches about 19%, while models incorporating additional energy fluxes have probabilities between 8% and 15%. When compared to reference forecasts produced with both mean model parameters and parameters tuned on single mass balance observations, the mass balances produced with the particle filter performs about equally well on the daily scale, but outperforms predictions of cumulative mass balance. The particle filter improves the performance scores of the reference forecasts by 91-97% in these cases. A leave-one-out cross-validation on the individual glaciers shows that the particle filter is able to reproduce point observations at locations on the glacier where it was not calibrated, as the filtered mass balances do not deviate more than 8% from the cumulative observations at the test locations. A comparison with glacier-wide annual mass balance by Glacier Monitoring Switzerland (GLAMOS) involving additional measurements distributed over the entire glacier, mostly show good agreement, but also deviations of up to $0.41\,\mathrm{m\,w.e.}$ for one instance.





## 1 Introduction

Glaciers around the world are shrinking. For example, Switzerland hast lost already more than a third of its glacier volume since the 1970s (Fischer et al., 2015), glaciers are currently melting at about $-0.6\,\mathrm{m\,w.e.a^{-1}}$ on average (Sommer et al., 2020), and it is expected that glaciers will continue to lose mass (Jouvet et al., 2011; Salzmann et al., 2012; Beniston et al., 2018; Zekollari et al., 2019). Since they fulfil important functions like water supply for drinking, irrigation and electricity production, there is high interest in near real-time glacier mass balance information. The near real-time mass balance status of glaciers in a summer has become important to public outreach in recent years for demonstrating the consequences of climate change (e.g. Euronews, 2019; Science Magazine, 2019).

A glacier mass balance nowcasting framework with data assimilation of observations could deliver these near real-time mass balances whenever they are requested. While nowcasting frameworks exist e.g. for the mass balance of the Greenland Ice Sheet based on satellite information combined with modelling (Fettweis et al., 2013; NSIDC, 2020a), for snow (NSIDC, 2020b; SLF, 2020), or for hydrological purposes (Zappa et al., 2008; Pappenberger et al., 2016; Zappa et al., 2018; WSL, 2020; Hydrique, 2020; Wu et al., 2020), there are no specific frameworks that provide analyses at high frequency incorporating observations for mountain glaciers yet. In general, data assimilation is widespread in oceanography, meteorology, hydrology and snow sciences "but its introduction in glaciology is fairly recent" (Bonan et al., 2014). Especially regarding glacier mass balance studies, data assimilation and Bayesian approaches appear only slowly in published work (Dumont et al., 2012; Leclercq et al., 2017; Rounce et al., 2020; Werder et al., 2020).

In many cases, the most frequent mass balance analyses available are calculated twice a year and they are based on seasonal in situ observations (Cogley et al., 2011). This low observation frequency has several reasons. First, there are often no high-frequency in situ observations available to support data evaluation schemes and models, since these observations can only be acquired with a substantial effort in terms of time and manpower. Only recently, approaches to obtain high-frequency data with relatively low effort occur in the literature and include e.g. a point-based monitoring of melt or snow water equivalent on mountain glaciers (Hulth, 2010; Fausto et al., 2012; Keeler and Brugger, 2012; Biron and Rabatel, 2019; Carturan et al., 2019; Gugerli et al., 2019; Netto and Arigony-Neto, 2019). Second, near real-time estimates are often based on ensemble modelling, like in numerical weather forecasting. This is because near real-time estimates are often subject to high uncertainties related to the unknown current state of the atmosphere and model parameter uncertainties. Ensemble modelling is used in glaciology in the context of model intercomparison projects (Hock et al., 2019), future projections for ice sheets and mountain glaciers (Ritz et al., 2015; Shannon et al., 2019; Golledge, 2020; Marzeion et al., 2020; Seroussi et al., 2020), and also to determine the initial conditions for modelling (Eis et al., 2019). However, ensembles are not prominent in the calculation of seasonal glacier mass balances.

Third, there is often a lack of knowledge about the exact short-term parameters in mass balance models. This poses a problem, since e.g. temperature index (TI) models are parametrizations of the full energy balance equation and deliver inaccurate results when applied with inapt parameters for a specific location. It has been underlined that TI models have the ability to explain most of the mass balance variability (e.g. Ohmura, 2001), but due to a lack of data it is discussed how TI models



can be applied to short time scales (Lang and Braun, 1990; Hock, 2003; Hock et al., 2005). Gabbi et al. (2014) showed in a comparison of four TI models and a full energy balance model that all models perform very similarly on a multi-year scale.

In this study, we address the issue of low-frequency observations, ensemble modelling and lack of knowledge about short-term parameters as part of the project Cryospheric Monitoring and Prediction Online (CRAMPON), which aims at delivering near real-time glacier mass balance estimates for mountain glaciers using data assimilation. To obtain more high-frequency data at a relatively low cost, we equipped three Swiss glaciers – Glacier de la Plaine Morte, Findelgletscher and Rhonegletscher – with in total seven camera instrumentations in summer 2019. Each of these instrumentations takes images of a 2 cm-marked mass balance stake at 20 minute intervals, and can thus deliver estimates of surface point mass balance aggregated to the daily scale. We assimilate these observations into an ensemble of three TI models and one simplified energy balance model using a particle filter, since particle filters do not restrict the class of state transition models or observation error distributions (Arulampalam et al., 2002; Beven, 2009; Magnusson et al., 2017). By designing our particle filter so that each model has a minimum contribution to the mass balance model ensemble, we put a special effort in ensuring that the ensemble is stable and suitable for operational use. In particular, temporarily badly performing models are not excluded from the predictions, but can recover later. To address the parameter uncertainty issue in TI models, we drive the mass balance model ensemble with Monte Carlo samples of uncertain meteorological input and prior parameter distributions obtained from past calibration on seasonal mass balances. By using an augmented state formulation of the particle filter, we make use of the property of particle filters to constrain model parameters as well (e.g. Ruiz et al., 2013).

As a result, we demonstrate (1) how such a workflow including daily melt observations, ensemble modelling and data assimilation works in practice, (2) to which extent the assimilated mass balances are able to reproduce the cumulative observations, and (3) how the ensemble performs with respect to reference forecasts and seasonal, operational analyses from in-situ measurements.

## 2 Study sites, data, and field instrumentation

We use Glacier de la Plaine Morte, Rhonegletscher, and Findelgletscher in summer 2019 as test sites (Figure 1). The basic morphological characteristics and instrumentations of these glaciers are given in Table 1.

### 2.1 Continuous in-situ mass balance observations

#### 2.1.1 Technical camera station setup

For an automated reading of daily point mass balances in the field, we use off-the-shelf cameras and logger boxes from the company Holfuy Ltd. . We mount these to an aluminium stake construction that we designed for glacier applications. Figure 2 provides an overview of the camera installation.

The camera observes an ablation stake, which is marked with colored tape at 2 cm intervals. When the surface melts, the stake construction slides along the mass balance stake and the camera records a picture of the stake every 20 minutes. Pictures





**Figure 1.** Locations of the glaciers equipped with cameras within Switzerland (a), and detailed topographic maps of the glaciers with dots for camera (red) and reference mass balance stake (blue) locations (b-d). All coordinates are given as Swiss Coordinates (EPSG:21781), the blue glacier outlines stem from GLAMOS, and background web mapping service tiles are provided by ©swisstopo/ ©Google Maps.

are sent in real time to our servers via the Swiss mobile phone network. Like this, we are able to obtain daily read-outs of glacier surface height change relative to the stake top as the basis for ablation measurements (Cogley et al., 2011). All pieces of the construction are lightweight (4 kg for the station + 4 kg for 8 m of mass balance stakes) and can be mounted by one person.



**Table 1.** Morphological features and camera settings of the investigated glaciers. Area and elevation refer to the year 2019 (GLAMOS, 2020), slope and aspect have been calculated using a recent Digital Elevation Model (DEM) (swisstopo, 2020)

| Parameter | Glacier de la Plaine Morte | Findelgletscher | Rhonegletscher |
|---|---|---|---|
| Area (km$^2$) | 7.1 | 12.7 | 15.3 |
| Elevation Range (m.a.s.l.) | 2470-2828 | 2561-3937 | 2223-3596 |
| Average Slope (°) | 6 | 13 | 14 |
| Average Aspect (°) | 341 (NNW) | 321 (NW) | 225 (SW) |
| Camera Stations | PLM 1 (2681 m.a.s.l.) | FIN 1 (2564 m.a.s.l.) FIN 2 (3021 m.a.s.l.) | RHO 1 (2233 m.a.s.l.) RHO 2 (2235 m.a.s.l.) RHO 3 (2392 m.a.s.l.) RHO 4 (2589 m.a.s.l.) |

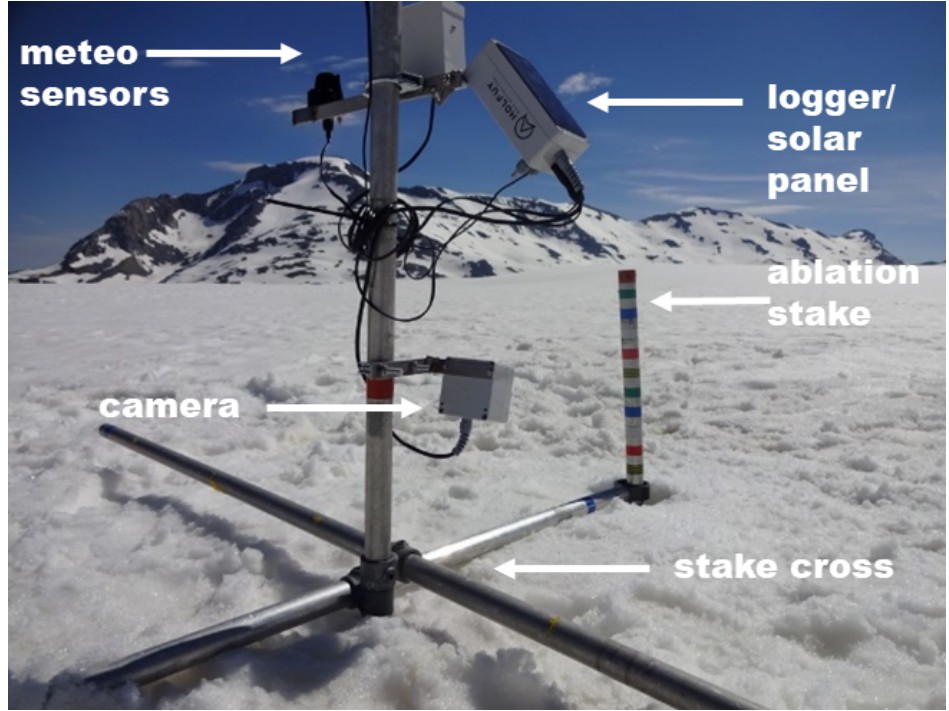

**Figure 2.** The camera construction used to obtain daily estimates of glacier point mass balance. Here, the camera has just been mounted on the snow covered surface of Glacier de la Plaine Morte (June 19th, 2019).





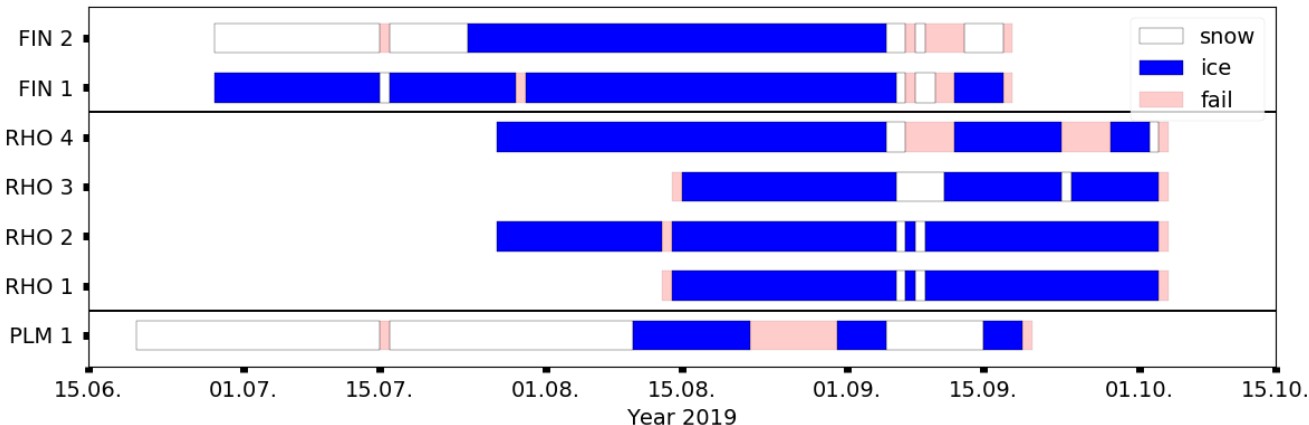

**Figure 3.** Overview of camera station availability during summer 2019. Cameras have been mounted and torn down at different times due to weather and staff restrictions. Station names on the y axis are defined as in Table 1. The category "snow" means that the glacier surface is snow covered, "ice" stands for bare ice exposed, and "fail" indicates either a failure in image transmission, station maintenance or the inability to read the mass balance.

### 2.1.2 Camera acquisitions in summer 2019

Figure 3 shows an overview of camera acquisitions and data gaps over the summer 2019. In total, we were able to obtain 352 daily point mass balance observations between June 20th, 2019 and October 3rd, 2019. The camera longest in the field was on Glacier de la Plaine Morte (91 days between June 20th, 2019 and September 18th, 2019), while those shortest in the field were two cameras at the tongue of Rhonegletscher (52 days between August 13th, 2019 and October 2nd, 2019). Very few data gaps occurred due to failure of the mobile network over which the data were transmitted.

Once camera images are on our servers, they are read manually to obtain daily cumulative surface height change $h(t,z)$ since camera setup. We assume that the observational error $\epsilon_t$ of a reading is Gaussian distributed and uncorrelated in time and space. To estimate the standard deviation of the Gaussian error distribution, we performed a round robin experiment with seven participants. In this kind of experiment all participants are given the task to read the same camera mass balances independently, and statistics are made about the degree of agreement between the individual assessments. Here, we found a standard deviation of 1.5 cm w.e. (range from 0.2 cm w.e. to 1.7 cm w.e.). This accounts for reading errors, errors in stake marker positions and unknown thickness of the melt crust on the ice surface, but this does not account for systematic errors.

The relationship between observations of cumulative ice surface height change between two time steps and the cumulative glacier mass balance is given through the simple linear observation operator $\mathcal{H}$:

$$h(t,z) = \mathcal{H}(b_{\mathrm{sfc}}(t,z), \epsilon_{t,z}) = \frac{b_{\mathrm{sfc}}(t,z) \cdot \rho_w}{\rho_{\mathrm{ice}}} + \epsilon_t \tag{1}$$

where $b_{\mathrm{sfc}}(t,z)$ (m w.e.) is the accumulated surface mass change at elevation $z$ and time $t$ since the day of the first camera observation, $\rho_w = 1000 \ \mathrm{kg\,m^{-3}}$ is the water density, and $\rho_{\mathrm{ice}}$ is the ice density (we assume $900 \ \mathrm{kg\,m^{-3}}$). To avoid systematic



errors of the mass balance readings, we exclude the initial snow-covered phase after camera setup at the stations FIN 2 and PLM 1. This is mainly because it can happen that the camera construction sinks into the snow cover and makes the daily

snow melt signal temporarily biased. This "sinking bias" is in most cases impossible to distinguish from the actual melt signal. Moreover, the short-term density of melting snow is unknown. Short snow events during the melt seasons have been assigned an estimated snow-water equivalent with high uncertainty though. This is done mainly to retain some information on the accumulation conditions at all. If a stake reading was impossible, we have resumed with a zero balance after the snow had melted again. However, there are also three cases that require special attention when reading mass balances on days without

snow: (1) maintenance operations like setup, redrilling and unmounting of a station, (2) melt that happened during night and is thus only visible on the next day, and (3) data gaps. Regarding maintenance operations, we do not consider the observations from days when maintenance has taken place. This is because those days are either not fully covered, or the mass balance stake and the entire station might melt into the ice after redrilling. For melt during night, we equally distribute the overnight melt between the two concerning days as a trade-off between warmer temperatures before midnight and colder temperatures

but longer time span after midnight. For data gaps, we have experienced only short image transmission outages which were mainly due to a six-day failure in the mobile network connection on Plaine Morte during September 2019. We have excluded the daily readings on these days, but we were able to reconstruct estimates of cumulative mass balance over the gap time span when acquisitions had resumed.

## 2.2 Meteorological input data

To model glacier mass balance, we employ verified products of daily mean and maximum $2\,\mathrm{m}$ temperature $T$ and $T_{\mathrm{max}}$, precipitation sum $P$ and mean incoming shortwave radiation $G$ from MeteoSwiss as model input (MeteoSwiss, 2017, 2018, 2019). These are delivered on grids with approx. $0.2°$ spatial resolution.

Temperature uncertainty, given as a root-mean-square error, varies per season from $0.94\,\mathrm{K}$ (MAM) to $1.67\,\mathrm{K}$ (DJF) in the Alpine region (Frei, 2014, 2020). We assume a Gaussian distributed additive error, which is perfectly spatially correlated for

a single glacier, but independent on different days. The perfect error correlation assumption can be justified with the fact that the station network from which the gridded temperature values are interpolated is much sparser than the scale of individual glaciers. The air temperature gradient is derived from a linear regression of the 25 closest cells to a glacier outline centroid. If the regression is not significant, we assign a lapse rate of $-0.0065\,\mathrm{K\,m^{-1}}$.

For operational reasons, the precipitation grids contain the 06 am - 06 am local time precipitation sums and are thus not

conform with the 00 am - 00 am temperature values (MeteoSwiss, 2019; Isotta et al., 2019). This might introduce an error, which we cannot account for though. Like for temperature, we thus focus again on random errors and pretend for simplicity that the precipitation sum was also from 00 am - 00 am. Precipitation uncertainty is generally harder to assess than temperature uncertainty, since it involves undercatch errors and skew error distributions. Here, we follow an error assessment by quantiles of precipitation intensity (Isotta et al., 2014; Frei, 2020). This assessment states that for the Alpine region the standard error at

moderate precipitation intensities is roughly an over-/underestimation factor of 1.25. The error increases (decreases) towards low (high) precipitation intensities, and it is generally slightly higher in the summertime. We draw samples from a multiplica-



tive Gaussian error distribution and, for same reason as for temperature, we assume perfect precipitation error correlation at the glacier scale. To achieve consistency with the temperature processing, we also derive precipitation lapse rates from the surrounding 25 grid cells. If the correlation is not significant, a rate of +0.02% $\mathrm{m}^{-1}$ is assigned as a compromise (e.g. Farinotti
et al., 2012; Schäppi, 2013; Huss and Fischer, 2016).

Shortwave radiation data are derived using data from the geostationary satellite series Meteosat. As an uncertainty, Stöckli (2013) gives a mean absolute bias between 9 and 29 $\mathrm{W\,m}^{-2}$. We assume the errors to be Gaussian and assign a standard deviation of 15 $\mathrm{W\,m}^{-2}$, perfectly correlated on the glacier scale and independent in time. Shortwave radiation is downscaled from the grid to the glacier with potential radiation (Section 3.1).

## 2.3 Glacier outlines and measured mass balances

Glacier outlines for the year 2019 are obtained from GLAMOS, and mass balances in this study are calculated using these outlines as a reference surface (Elsberg et al., 2001; Huss et al., 2012).

For calibration and verification, we use different mass balance data which are acquired in the frame of GLAMOS (Glacier Monitoring Switzerland, 2018). First, intermediate readings of stakes independent from the near real-time stations but nearby
our installations have been made explicitly for this study. Stake locations are depicted in Figure 1. The reading error for these measurements is usually estimated to be around 5 cm (e.g. Müller and Kappenberger, 1991). Second, we use glacier-wide seasonal mass balances that are based on in-situ observations covering the glacier surface acquired during two field campaigns in April and September, respectively. Values of glacier-wide mass balance are obtained by extrapolating measurements and are partly harmonized with long-term geodetic mass balances (e.g. Bauder et al., 2007; Huss et al., 2015). The extrapolation method
to infer glacier-wide mass balance from point measurements involves an adjustment of model parameters of an accumulation and TI melt model (Hock, 1999) at locations where observations are available, while mass balances at grid cells without observations are produced using the calibrated model (Huss et al., 2009, 2015). Uncertainties of the glacier-wide annual mass balance for the measurement period have been estimated to be 0.09-0.2 m w.e. in six experiments where GLAMOS (1) model parameters (temperature lapse rate, ratio between melt coefficients, summer precipitation correction) and (2) snow extrapolation
parameters have been varied within prescribed ranges, and (3) mass balance stake reading uncertainty, (4) DEM and outline uncertainty, (5) climate forcing uncertainty and (6) point data availability have been accounted for.

## 3 Methods

### 3.1 Mass balance modelling

Glacier surface mass balance consists of two components: accumulation and ablation. We model accumulation and ablation
on elevation bins whose vertical extent is determined by a $\approx 20$ m horizontal spacing of nodes along the central flow line of the glacier, which serve as mean height of an elevation band (Maussion et al., 2019). To obtain glacier-wide mass balance, node mass balances are weighted with the area per elevation bin. To compute accumulation at different elevations, we employ



a simple, but widely used accumulation model (e.g. Huss et al., 2008):

$$c_{\mathrm{sfc}}(t,z) = c_{\mathrm{prec}}(t) \cdot P_{\mathrm{s}}(t) \cdot [1 + (z - z_{\mathrm{ref}}) \cdot \frac{\partial P_{\mathrm{s}}}{\partial z}], \tag{2}$$

where $c_{\mathrm{sfc}}(t,z)$ (m w.e.) is the snow accumulation at time step $t$ and elevation $z$, $c_{\mathrm{prec}}(t)$ is the unitless multiplicative precipitation correction factor, $P_{\mathrm{s}}(t)$ is the sum of solid precipitation at the elevation of the precipitation reference cell $z_{\mathrm{ref}}$ and time step $t$, and $\frac{\partial P_{\mathrm{s}}}{\partial z}$ is the solid precipitation lapse rate. Following Sevruk (1985), we choose $c_{\mathrm{prec}}$ to vary sinusoidally by $\pm 8\%$ around its mean during one year, being highest in winter and lowest in summer. This is to account for systematic average variations in gauge undercatch depending on the precipitation phase. The water phase change in the temperature range around 0 °C is
modeled using a linear function between 0 °C and 2 °C, i.e. at 1°C there is 50% snow and 50% rain (e.g. Maussion et al., 2019).

  Since all three glaciers we investigate are in the GLAMOS measurement program and winter mass balance observations are available, the effect of spatial variations in snow accumulation, differing from a linear gradient, can be incorporated: by adjusting a factor $D(z)$ the model mass balance in the elevation bins is altered such that it matches the interpolated distribution
of measured winter mass balances (Farinotti et al., 2010):

$$C_{\mathrm{sfc,\,glamos}}^{w}(z) = D(z) \cdot C_{\mathrm{sfc}}^{w}(z) \tag{3}$$

with $C_{\mathrm{sfc}}^{w}(z)$ (m w.e.) being the modelled winter surface accumulation, i.e. the sum of individual $c_{\mathrm{sfc}}(t,z)$ over the winter period, and $C_{\mathrm{sfc,\,glamos}}^{w}(z)$ (m w.e.) being the interpolated GLAMOS winter surface accumulation measurements at the individual elevation bins $z$.

To model surface ablation, we set up an ensemble of three TI melt models and one simplified energy-balance melt model. We choose these individual ensemble models since they differ in the degree of complexity they use to describe the surface energy balance (Hock, 2003). They reach from using only temperature as input for determining melt via employing additionally the potential irradiation to using temperature and the actual short wave radiation. The ensemble contains

  1. the "BraithwaiteModel" using only air temperature as input to calculate melt (Braithwaite and Olesen, 1989; Braithwaite,
195   1995):

$$a_{\mathrm{sfc}}(t,z) = \mathrm{DDF}_{\mathrm{snow/ice}} \cdot \max(T(t,z) - T_{\mathrm{melt}},\,0) \tag{4}$$

   where $a_{\mathrm{sfc}}(t,z)$ (m w.e. $\mathrm{d}^{-1}$) and $T(t,z)$ (°C) are surface ablation and air temperature at time step $t$ and elevation $z$, respectively, $\mathrm{DDF}_{\mathrm{snow/ice}}$ (m w.e. $\mathrm{K}^{-1}\,\mathrm{d}^{-1}$) are the temperature sensitivities ("degree-day factors") of the surface types (snow/ice), $\max()$ is the maximum operator, and $T_{\mathrm{melt}}$ (°C) is the threshold temperature for melt. For this application,
we set $T_{\mathrm{melt}}$ to 0 °C and keep the ratio of $\mathrm{DDF}_{\mathrm{snow}}/\mathrm{DDF}_{\mathrm{ice}}$ constant at 0.5 (Hock, 2003).

  2. the "HockModel" using potential incoming solar radiation as an additional predictor for melt (Hock, 1999):

$$a_{\mathrm{sfc}}(t,z) = (\mathrm{MF} + a_{\mathrm{snow/ice}} \cdot I_{\mathrm{pot}}(t,z)) \cdot \max(T(t,z) - T_{\mathrm{melt}}, 0) \tag{5}$$





where MF ($\mathrm{m\,w.e.\,K^{-1}\,d^{-1}}$) is the temperature melt factor, $a_{\mathrm{snow/ice}}$ ($\mathrm{m\,w.e.\,m^2\,d^{-1}\,W^{-1}\,K^{-1}}$) are the radiation coefficients for snow and ice, respectively, $I_{\mathrm{pot}}(t,z)$ ($\mathrm{W\,m^{-2}}$) is the potential clear-sky direct solar radiation at time $t$ and elevation $z$, $T_{\mathrm{melt}}$ is set again to $0\,^\circ\mathrm{C}$ and the ratio of $a_{\mathrm{snow}}/a_{\mathrm{ice}}$ is 0.8 (Hock, 1999; Farinotti et al., 2012). $I_{\mathrm{pot}}(t,z)$ is computed at ten minute intervals following the methods described in Iqbal (1983), Hock (1999) and Corripio (2003), and by using swissALTI3D (swisstopo, 2020) as a background elevation model. Daily values are then obtained by averaging, and values for the different glacier elevations are aggregated. We assume equal uncertainties for both actual and potential incoming shortwave radiation $G$ and $I_{\mathrm{pot}}$.

3. the "PellicciottiModel" employing explicit surface albedo and actual incoming short-wave solar radiation (Pellicciotti et al., 2005):

$$a_{\mathrm{sfc}}(t,z) = \begin{cases} \mathrm{TF} \cdot T(t,z) + \mathrm{SRF} \cdot (1 - \alpha(t,z)) \cdot G(t,z), & \text{for } T(t,z) > T_{\mathrm{melt}} \\ 0, & \text{for } T(t,z) \le T_{\mathrm{melt}} \end{cases} \tag{6}$$

where TF ($\mathrm{m\,w.e.\,K^{-1}\,d^{-1}}$) is the temperature factor, SRF ($\mathrm{m^3\,d^{-1}\,W^{-1}}$) is the shortwave radiation factor, and $\alpha(t,z)$ and $G(t,z)$ ($\mathrm{W\,m^{-2}}$) are the albedo and incoming shortwave radiation at time $t$ and elevation $z$, respectively. Note that in this case $T_{\mathrm{melt}}$ is equal to $1\,^\circ\mathrm{C}$ (Pellicciotti et al., 2005).

Albedo is approximated according to the combined decay equation for deep and shallow snow in Brock et al. (2000):

$$\alpha(t,z) = (1 - e^{(-\mathrm{swe}(t,z)/\mathrm{swe}^*)}) \cdot (p_1 - p_2 \cdot \log_{10}(T_{\mathrm{acc}}(t,z))) + e^{(-\mathrm{swe}(t,z)/\mathrm{swe}^*)} \cdot (\alpha_u(t,z) + p_3 \cdot e^{-p_4 \cdot T_{\mathrm{acc}}(t,z)}) \tag{7}$$

where $\mathrm{swe}(t,z)$ is the snow water equivalent at time $t$ and elevation $z$, $\mathrm{swe}^* = 0.024\,\mathrm{m\,w.e.}$ is a scaling length for swe, $p_1 = 0.713$, $p_2 = 0.155$, $p_3 = 0.442$ and $p_4 = 0.058$ are empirical coefficients as given in Brock et al. (2000), $\alpha_u$ is the albedo of the underlying firn/ice below the snow, and $T_{\mathrm{acc}}(t,z)$ is the accumulated daily maximum temperature $> 0\,^\circ\mathrm{C}$ since a snowfall event at elevation $z$. To avoid infeasible albedo values, $\alpha(t,z)$ is clipped as suggested in Brock et al. (2000).

4. the "OerlemansModel" calculating melt energy as the residual term of a simplified surface energy balance equation (Oerlemans, 2001):

$$a_{\mathrm{sfc}}(t,z) = \frac{Q_{\mathrm{m}}(t,z)\,dt}{L_f\,\rho_w} \tag{8}$$

where

$$Q_{\mathrm{m}}(t,z) = (1 - \alpha(t,z)) \cdot G(t,z) + c_0 + c_1 \cdot T(t,z). \tag{9}$$

In the above equations, $Q_{\mathrm{m}}(t,z)$ ($\mathrm{W\,m^{-2}}$) is the melt energy at time $t$ and elevation $z$, $dt = 1$ day is a time step, $L_f = 3.34 \cdot 10^5$ ($\mathrm{J\,kg^{-1}}$) is the latent heat of fusion, and $c_0$ ($\mathrm{W\,m^{-2}}$) and $c_1$ ($\mathrm{W\,m^{-2}\,K^{-1}}$) are empirical factors. Albedo is calculated as well according to Equation (7).



## 3.2 Mass balance model calibration

For the data assimilation procedure described in Section 3.3, we need a prior estimate for the model parameter values of the mass balance equations (2), (4), (5), (6) and (9). This is why we calibrate all three investigated glaciers on the GLAMOS glacier-wide mass balances between mid of the 2000s and 2018 introduced in Section 2.3. To do this, we use an iterative

procedure similar to Huss et al. (2009) and illustrated in Figure 4. Additionally, we calibrate the snow redistribution factor $D(z)$ annually.

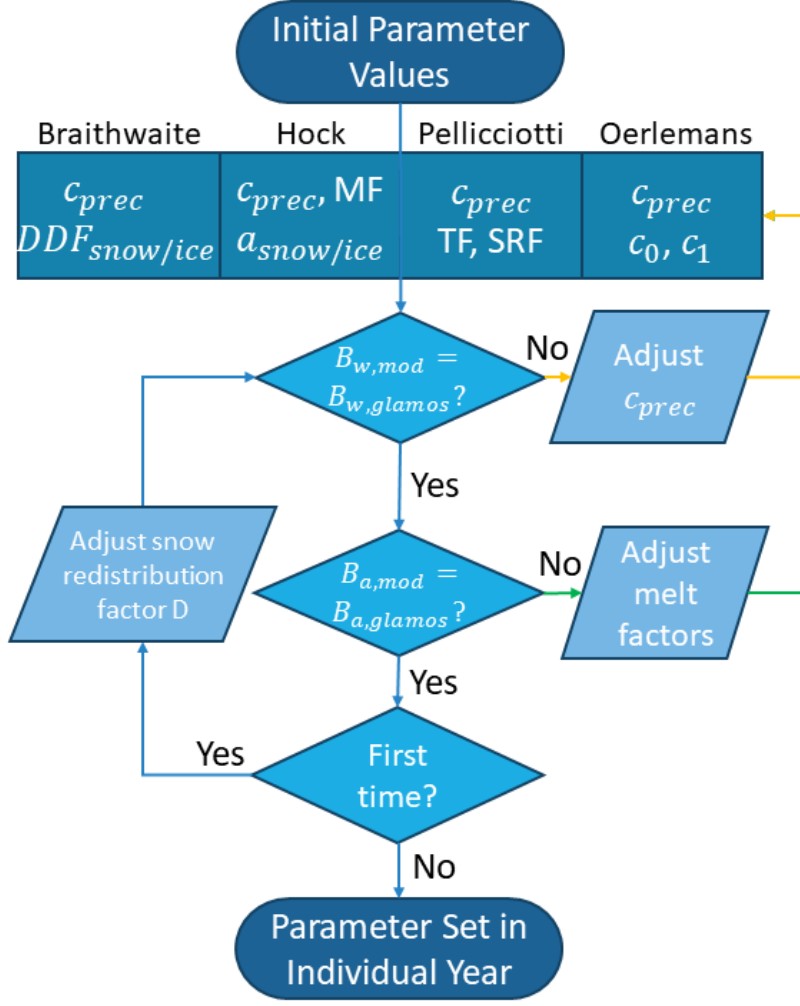

**Figure 4.** The calibration workflow. $B_{w,\,mod}$ and $B_{w,\,glamos}$ stand for glacier-wide modelled and GLAMOS winter mass balance, and $B_{a,\,mod}$ and $B_{a,\,glamos}$ stand for the modelled and GLAMOS annual mass balance, respectively. The yellow arrows highlight the first iteration step, while the green arrows highlight the second iteration step. Figure altered from Huss et al. (2009).





For the Huss et al. (2009) procedure, all model parameters are initially set to typical value ranges reported in the literature (Hock, 1999; Oerlemans, 2001; Pellicciotti et al., 2005; Farinotti et al., 2012; Gabbi et al., 2014), and then two calibration procedures are applied alternately: first, the precipitation correction factor is tuned so that the winter mass balance of a given year

is reproduced and the melt factors are held constant at their initial values. In a second step, the calibrated precipitation factor is kept constant, and the melt factors are optimized to reproduce the annual mass balance. Both precipitation correction and melt factors converge with every iteration. We terminate the iteration procedure after the absolute difference to the winter/annual mass balance drops below 1 millimeter w.e..

Once optimized model parameters have been found, we calculate the value of $D(z)$ that matches the interpolated winter

mass balance. This may result in changes of the required model parameters. Therefore, as a final step, the iterative procedure is applied once more.

### 3.3   Particle filtering

To ensure that all mass balance model predictions stay within the observational uncertainty at every point in time, we perform data assimilation. In particular, we employ a particle filter, since it does not restrict the class of state transition models and

error distributions. Some extensions of the common particle filter framework allow estimating model parameters and model performance over time. With this, we would like to give optimal, daily mass balance estimates at the glacier scale.

### 3.3.1   General framework

The general framework for data assimilation consists of a system whose state $\boldsymbol{x}_t$ evolves according to a model, but only partial and uncertain observations $\boldsymbol{y}_t$ of the state are available:

$$\begin{cases} \boldsymbol{x}_t = g(\boldsymbol{x}_{t-1}, \boldsymbol{\beta}_t), & \text{state transition equation} \\ \boldsymbol{y}_t = \mathcal{H}(\boldsymbol{x}_t) + \boldsymbol{\epsilon}_t & \text{observation equation} \end{cases} \tag{10}$$

Here, $\boldsymbol{x}_{t-1}$ is the state at the previous time step, $g(\cdot)$ is the state transition function, $\mathcal{H}$ is the observation operator as introduced in Equation (1), $\boldsymbol{\epsilon}_t$ is the observation error vector at time $t$, and $\boldsymbol{\beta}_t$ is a random variable that describes model uncertainties. The term for $\boldsymbol{\beta}_t$ does not need to be strictly additive, and it can also represent uncertainties in model input variables. The goal of data assimilation is to compute conditional distributions of the system state $\boldsymbol{x}_t$ based on observations $\boldsymbol{y}_{1:t} = (\boldsymbol{y}_1, \boldsymbol{y}_2, ..., \boldsymbol{y}_t)$

sequentially for $t = t_0, t_0 + 1, ... t_{\text{end}}$, where $t_0$ and $t_{\text{end}}$ are the time steps with the first and last observations, respectively. In our case, these conditional distributions describe the cumulative mass balance state of a glacier, given all available camera observations.

To put this general framework into practice, we use the particle filter, which is a sequential Monte Carlo data assimilation method. Instead of handling conditional distributions of $\boldsymbol{x}_t$ analytically, the particle filter approximates the conditional distri-

bution of a state $\boldsymbol{x}_t$ at time $t$ given the observations $\boldsymbol{y}_{1:t}$ by a weighted sample of size $N_{\text{tot}}$ (e.g. van Leeuwen et al., 2019):





$$p(\boldsymbol{x}_t \mid \boldsymbol{y}_{1:t}) \approx \sum_{k=1}^{N_{\text{tot}}} w_{t,k}\delta(\boldsymbol{x}_t - \boldsymbol{x}_{t,k}), \quad \sum_{k=1}^{N_{\text{tot}}} w_{t,k} = 1 \tag{11}$$

Here $p(\cdot)$ means "probability of", $\delta(\cdot)$ is the Dirac Delta function, the elements $\boldsymbol{x}_{t,k}$ of the sample are called "particles" and the weights $w_{t,k}$ associated with the particles $\boldsymbol{x}_{t,k}$ sum to unity.

Usually, particle filtering comprises three repeated steps: the predict step, the update step, and the resampling step. In our case, these steps mean the following: During the predict step, particles holding possible mass balance states are propagated forward in time using the state transition equation in Equation (10), where $g(\cdot)$ represents the ensemble prediction of mass balance equations (2) - (7). This acts as a prior estimate of the mass balance distribution. In the update step, the weights of the propagated particles are recalculated based on Bayes' theorem. This accounts for the information of the next camera mass

balance observation. In the last step, particles are resampled according to the updated weights. This step is necessary to restore particle diversity that is reduced during the update step. Resampling avoids so-called particle degeneracy, where all weights collapse on only a few particles. Beyond the common three-step scheme, we additionally estimate model parameters with the particle filter by augmenting the state vector with model parameter values. In this way, we add an additional fourth step to the particle filter scheme, where we evolve model parameters temporally according to a defined memory parameter. This prevents

a collapse of the ensemble due to overconfidence, meaning that model parameter variability would become too low over time.

### 3.3.2 Application of the framework

The flowchart in Figure 5 visualizes how the particle filter is implemented in our mass balance modeling framework. Figure 6 sheds light on how we perform the individual particle filter steps.

The temporal dynamics of the glacier mass balance state can be described by the accumulation model in Equation (2)

combined with the four different melt models in Equations (4), (5), (6), (8), and it is not known which model performs best. In addition, each model has its own unknown parameters. To take these uncertainties into account, we augment the state vector by the model index $m_t \in \{1, 2, 3, 4\}$ and the model parameters $\boldsymbol{\theta}_t$. In this way, a model and its parameter values are also estimated based on the observations. Although the unknown parameters are different for each model, we do not use an additional model index for $\boldsymbol{\theta}_t$. Instead, we ensure that for all particles, $\boldsymbol{\theta}_{t,k}$ is always the parameter vector associated with model $m_{t,k}$.

As the state has to provide all information that is needed to predict the next observation, we also include the surface albedo and the snow water equivalent on the ice in our state vector. Hence it is defined as

$$\boldsymbol{x}_t = (m_t, \boldsymbol{\theta}_t, \boldsymbol{\xi}_t), \quad \boldsymbol{\xi}_t = (b_{\text{sfc}}(t,z), \alpha(t,z), \text{swe}(t,z)) \tag{12}$$

where we call $\boldsymbol{\xi}_t$ the physical state.

### 3.3.3 Predict step

During the predict step, the explicit temporal evolution of the physical state $\boldsymbol{\xi}_t$ involves the randomized error draws which account for uncertainties in the meteorological input variables (Section 2.2). Here, we call them $\boldsymbol{\eta}_t$ and set an additional



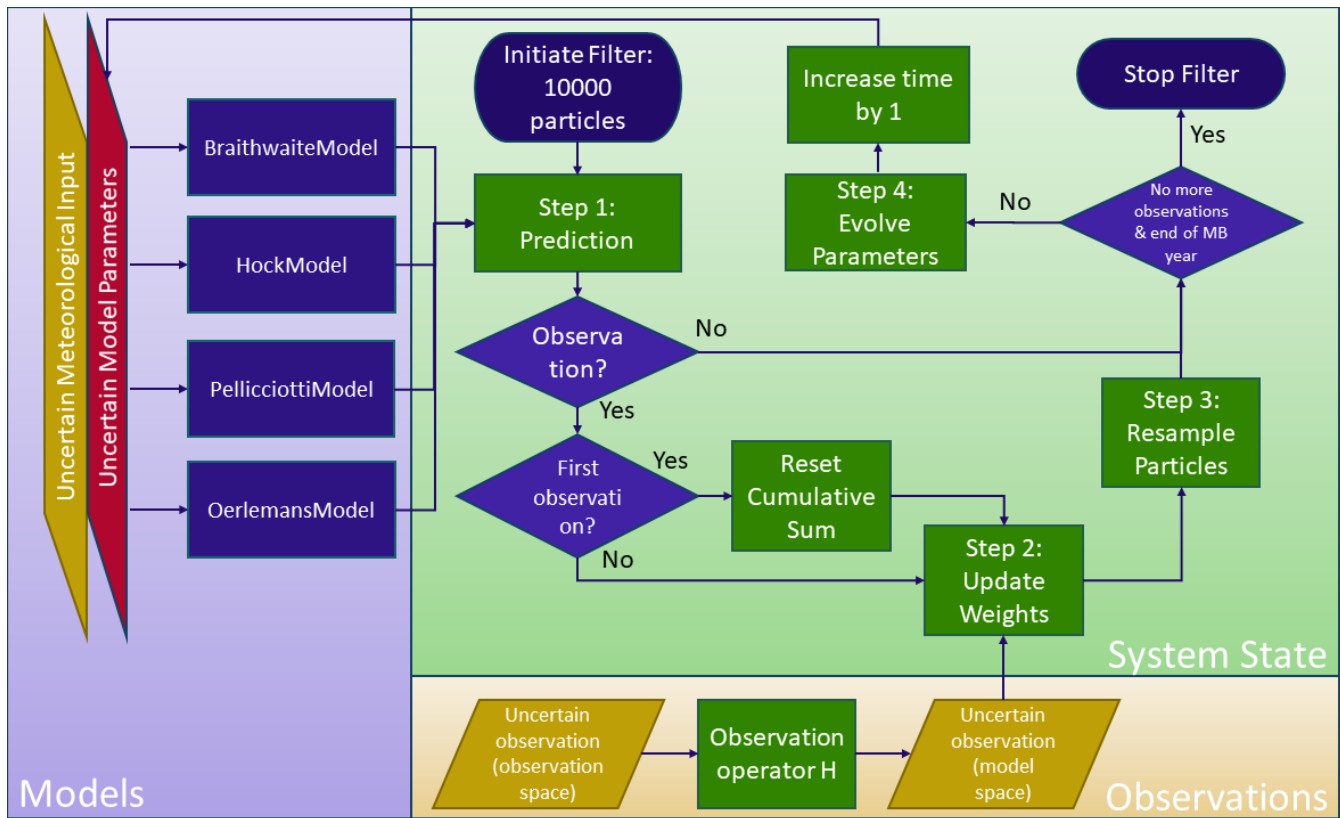

**Figure 5.** Particle filter workflow during one mass budget year ("MB year"). We use uncertain model estimates to predict mass balance with 10000 particles and reset the cumulative mass balance when a camera is set up. The model mass balance estimate is updated at time steps where observations are available. To avoid overconfidence of the particle filter, we apply a partial resampling technique.

scalar subscript to indicate that the errors are different for each meteorological variable. As by Equation (7) the new albedo is determined by the accumulated daily maximum temperature $T_{\mathrm{acc}}$ since a snowfall event and $\mathrm{swe}(t,z)$ determines the melt factor in Equations (4) and (5), we first predict $c_{\mathrm{sfc}}(t,z)$, $T_{\mathrm{acc}}(t,z)$, and $\mathrm{swe}(t,z)$:

$$c_{\mathrm{sfc}}(t,z) = c_{\mathrm{sfc}}(P_{\mathrm{s}}(t,z), \eta_{t,2}, \boldsymbol{\theta}_t) \tag{13}$$

$$T_{\mathrm{acc}}(t,z) = \begin{cases} T_{\mathrm{acc}}(\alpha(t-1,z)) + T_{\mathrm{max}}(t,z) + \eta_{t,1}, & \text{if } T_{\mathrm{max}}(t,z) > 0 \text{ and } c_{\mathrm{sfc}}(t,z) < 0.001\,\mathrm{m\,w.e.\,d}^{-1} \\ 0, & \text{otherwise.} \end{cases} \tag{14}$$

$$\mathrm{swe}(t,z) = \max(\mathrm{swe}(t-1,z) - a_{\mathrm{sfc}}(t-1,z), 0) + c_{\mathrm{sfc}}(t,z) \tag{15}$$





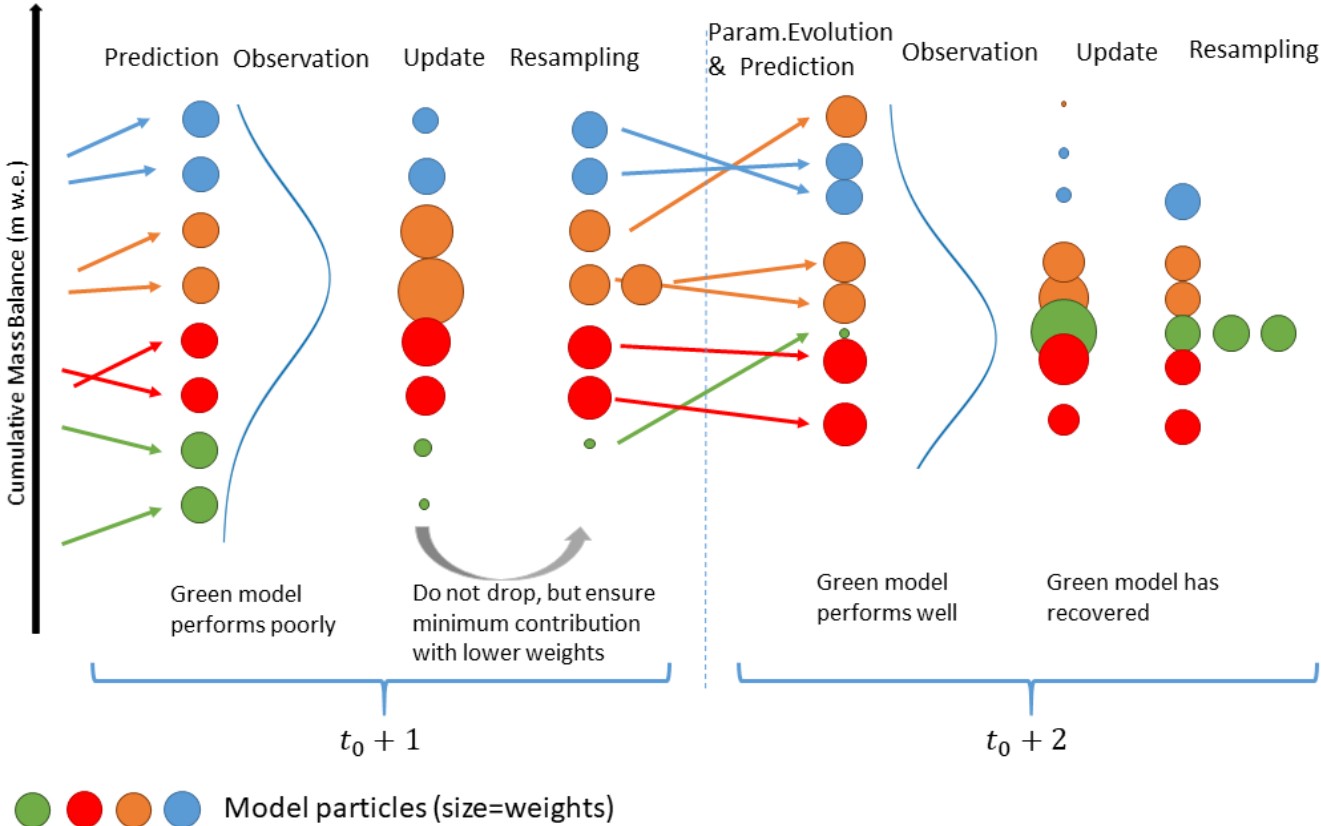

**Figure 6.** Illustration of the individual particle filter steps. The example refers to a case in which four models (blue, orange, red, and green) start with two particles each. The blue curve represents the observation distribution. At time step $t_0 + 1$, the green model performs poorly and receives entirely low weights during the update step (weights are shown by the size of the circles). In the resampling step, we modify the weights of the other particles again such that their weights compensate for not omitting the green model due to poor performance. As the green model stays in the ensemble, it can recover any time when making a good prediction (here: $t_0 + 2$).

Based on Equations (4), (5), (6) and (8), the predicted mass balance is then:

$$
\begin{aligned}
b_{\mathrm{sfc}}(t, z) = {} & b_{\mathrm{sfc}}(t - 1, z) \\
& + c_{\mathrm{sfc}}(t, z) \\
& - a_{\mathrm{sfc}}(T(t, z) + \eta_{t,3},\ G(t, z) + \eta_{t,4},\ \alpha(T_{\mathrm{acc}}(t, z)),\ \mathrm{swe}(t, z),\ m_t,\ \boldsymbol{\theta}_t) \\
& + \beta_t
\end{aligned}
\tag{16}
$$

where the errors $\boldsymbol{\eta}_t$ of the input variables shall be independent in time, but partly perfectly correlated in space for reasons described in Section 2.2. Since we already consider both parameter and input variable uncertainty, we set $\beta_t$ to zero for simplicity. Finally, the observations $\boldsymbol{y}_t$ depend only on the cumulative mass balance at the elevation $z$ of the camera as specified in Equation (1).





We use a total of $N_{\text{tot}} = 10,000$ particles and set the weights at starting time $t_0$, when a first camera observation is available,
to $1/N_{\text{tot}}$. Since at $t_0$ all models have equal probabilities, $N_{\text{tot}}/4$ particles are assigned to each of the four models. The initial
value of $b_{\text{sfc}}(t_0, z)$ is set to zero for all particles, whereas $\alpha(t_0, z)$ is determined by maximum air temperature values in the
meteorological data since the last snowfall before $t_0$, and $\text{swe}(t_0, z)$ depends on the cumulative mass balance before $t_0$. Finally,
the initial calibration parameter values $\boldsymbol{\theta}_{t_0,k}$ of the particles with model index $j$ are obtained by drawing Monte Carlo samples
from a normal distribution for the logarithmized parameter distribution of model $j$, calibrated in the past (see Section 3.2).
Table 2 shows the input parameter means and standard deviations for the three glaciers.

**Table 2.** Sample mean and covariance for the parameter prior distributions used on Glacier de la Plaine Morte, Findelgletscher and Rhone-gletscher.

| Parameter | Unit | Plaine Morte | Findel | Rhone |
|---|---|---|---|---|
| $\text{DDF}_{\text{ice}}$ | $\text{mm w.e. K}^{-1}\,\text{d}^{-1}$ | $6.81 \pm 0.87$ | $11.44 \pm 1.76$ | $8.53 \pm 0.84$ |
| MF | $\text{mm w.e. K}^{-1}\,\text{d}^{-1}$ | $2.55 \pm 0.95$ | $1.77 \pm 0.05$ | $1.79 \pm 0.02$ |
| $a_{\text{ice}}$ | $\text{mm w.e. m}^2\,\text{d}^{-1}\,\text{W}^{-1}\,\text{K}^{-1}$ | $0.009 \pm 0.007$ | $0.030 \pm 0.006$ | $0.014 \pm 0.002$ |
| TF | $\text{mm w.e. K}^{-1}\,\text{d}^{-1}$ | $2.85 \pm 0.21$ | $4.30 \pm 1.52$ | $3.80 \pm 1.09$ |
| SRF | $\text{m}^3\,\text{d}^{-1}\,\text{W}^{-1}$ | $0.07 \pm 0.03$ | $0.17 \pm 0.22$ | $0.08 \pm 0.05$ |
| $c_0$ | $\text{W m}^{-2}$ | $-114.22 \pm 1.77$ | $-106.30 \pm 9.07$ | $-112.64 \pm 3.13$ |
| $c_1$ | $\text{W m}^{-2}\,\text{K}^{-1}$ | $12.86 \pm 1.54$ | $17.55 \pm 3.00$ | $14.58 \pm 1.91$ |
| $c_{\text{prec}}$ | - | $1.60 \pm 0.20$ | $1.43 \pm 0.20$ | $1.56 \pm 0.25$ |

### 3.3.4 Update step

In the update step, all particles are then reweighted by multiplying the density of the observations $\boldsymbol{y}_t$ given the state of individual
particles $\boldsymbol{x}_{t,k}$ with their respective weights at $t - 1$ and normalizing the weights to sum to unity (van Leeuwen et al., 2019):

$$w_{t,k} = w_{t-1,k} \frac{p(\boldsymbol{y}_t \mid \boldsymbol{x}_{t,k})}{\sum_l w_{t-1,l}\, p(\boldsymbol{y}_t \mid \boldsymbol{x}_{t,l})} \tag{17}$$

Here, $p(\boldsymbol{y}_t \mid \boldsymbol{x}_{t,k})$ is the normal density with mean $b_{sfc}(t,z)_k/\rho_{\text{ice}}$ and variance $\sigma_\epsilon$ evaluated at $h(t,z)$. After updating the
model predictions with the observations, we are interested in (a) the posterior model probabilities $\pi_{t,j}$, (b) the posterior dis-
tribution of model parameters $\theta$, and of course (c) the posterior distribution of the physical state given all observations $\boldsymbol{y}_{1:t}$.
These quantities can be decomposed from the approximation with weighted particles in Equation (11). The posterior model
probability is given by

$$p(m_t = j \mid \boldsymbol{y}_{1:t}) \approx \pi_{t,j} = \sum_{k=1}^{N_{\text{tot}}} w_{t,k}\, \delta(m_{t,k} - j) \tag{18}$$





where $\pi_{t,j}$ is the approximation of the posterior model probability at time $t$ and model $j$. The posterior distribution of the parameters of model $j$ is approximated by

$$p(\boldsymbol{\theta}_t \mid \boldsymbol{y}_{1:t}, m_t = j) \approx \sum_{k=1}^{N_{\text{tot}}} \frac{w_{t,k}}{\pi_{t,j}} \delta(m_{t,k} - j) \delta(\boldsymbol{\theta}_{t,k} - \boldsymbol{\theta}_t). \tag{19}$$

The posterior distribution of the physical state takes the model uncertainty into account. It combines the posterior distributions
under the different models $j$ according to the law of total probability, where we can plug in Equations (18) and (19):

$$p(\boldsymbol{\xi}_t \mid \boldsymbol{y}_{1:t}) = \sum_{j=1}^{4} p(m_t = j \mid \boldsymbol{y}_{1:t}) p(\boldsymbol{\xi}_t \mid \boldsymbol{y}_{1:t}, m_t = j) \approx \sum_{j=1}^{4} \pi_{t,j} \sum_{k=1}^{N_{\text{tot}}} \frac{w_{t,k}}{\pi_{t,j}} \delta(m_{t,k} - j) \delta(\boldsymbol{\xi}_{t,k} - \boldsymbol{\xi}_t) = \sum_{k=1}^{N_{tot}} w_{t,k} \delta(\xi_{t,k} - \xi_t). \tag{20}$$

As the observations only measure the mass change since the installation of a camera, a difficulty occurs if several cameras are installed on different days at different elevations of the same glacier. We elaborate on the technical details for these cases in Appendix A.

### 3.3.5  Resampling

During the resampling step, the updated weights are used to choose a new set of $N_{\text{tot}}$ particles with equal weights. To achieve equal weights, particles with low weights are removed, whereas those with high weights are duplicated. Because there is no stochasticity in the evolution of $m_t$ though, for some models only a few particles with the according model index survive after a couple of iterations. If this occurs, the respective model has little chance to become better represented at later time steps,
which is unfavourable, since the model might give better predictions on average.

To overcome this problem, we choose a minimum model contribution to the ensemble, regardless of how poorly an individual model performs at a certain time step. To compensate for the potentially too high resampling rate of a poor prediction, we lower the weights of all particles of a model whose contribution has been deliberately increased to match the chosen minimum contribution. In turn, we increase the weights of all other particles to compensate for their underrepresentation, so that even-
tually the changed weights are equal to the original weights on average. For technical details of the resampling procedure see Appendix B.

### 3.3.6  Parameter Evolution

The dynamics of the augmented state is defined such that the model index does not change over time, but parameters are evolved temporally such that after a long period without observations $\boldsymbol{\theta}$ is distributed according to the prior parameter distribution:

$$\boldsymbol{\theta}_{t+1} = \rho \boldsymbol{\theta}_t + (1 - \rho) \boldsymbol{\mu}_0 + \boldsymbol{\zeta}_t, \quad \boldsymbol{\zeta}_t \sim \mathcal{N}(0, (1 - \rho^2) \boldsymbol{\Sigma}_0), \tag{21}$$

where $\boldsymbol{\mu}_0$ and $\boldsymbol{\Sigma}_0$ are the prior mean and the prior covariance of $\boldsymbol{\theta}$ at the starting time $t_0$ and $\rho \in [0; 1]$ is a memory parameter that we choose to be 0.9. This step accounts for the fact that parameters are not necessarily constant in time, and it also ensures to reintroduce parameter diversity which is lost during the resampling step.





### 3.4 Validation scores

To validate the daily mass balance prediction with the particle filter, we use the Continuous Ranked Probability Score (CRPS).
The CRPS is designed to estimate the deviation of a probabilistic forecast from an observation. The way it is constructed takes
into account both the deviation of the median forecast from the actual observation and the spread of the forecast distribution.
It is defined as (Hersbach, 2000):

$$\text{CRPS} = \int\limits_{-\infty}^{\infty} [P_f(b_\text{sfc}/\rho_\text{ice} \cdot \rho_w) - H(b_\text{sfc}/\rho_{ice} \cdot \rho_w - h(t,z))]^2 db_\text{sfc} \tag{22}$$

where $P_f(\cdot)$ is the forecast mass balance cumulative probability distribution, and $H(\cdot)$ is the Heaviside function. Lower
values of the CRPS correspond to better forecasts, and the minimum value is zero which corresponds to the deterministic
perfect forecast at the observation. The simplest choice for $P_f$ is the weighted ensemble distribution of the predict particles,
i.e. the discrete step function which has jumps of height $w_{t-1,k}$ at the positions $b_\text{sfc}(t,z)_k/\rho_\text{ice} \cdot \rho_w$ where $b_\text{sfc}(t,z)_k$ are the
particles from the predict step. There is a problem though, because this choice does not account for the observation error of
$h(t,z)$. This implies in particular that the score is not "proper", i.e. it does not always return the best value when the prediction
distribution is the true distribution (Ferro, 2017; Brehmer and Gneiting, 2019). To obtain a proper score one can use what
Ferro (2017) calls the error-convolved approach: instead of the discrete weighted ensemble forecast of the mass balance, this
approach uses the implied forecast of the camera reading $h(t,z)$, which is the Gaussian mixture with weights $w_{t-1,k}$ mean
values $b_\text{sfc}(t,z)_k/\rho_\text{ice} \cdot \rho_w$, and common variance $\sigma_\epsilon^2$. Despite being proper, this choice has still some disadvantages, because it
is not unbiased in the sense of Definition 3 of Ferro (2017). As for our data the values of the two scores do not differ much, we
use only the second choice in all results figures, but give also the value of the first choice in square brackets in the text.

## 4 Results and Discussion

### 4.1 Mass balance observations

Figure 7 shows the observed cumulative mass balance at the individual cameras, an example of meteorological conditions at
station FIN 1, daily mass balance rates at FIN 1, and four example camera images. We choose to show FIN 1, since it is the
longest observation time series of ice melt. Considering all stations, we have observed ice melt rates of up to $0.12 \, \text{m w.e.} \, \text{d}^{-1}$
and a cumulative mass balance of about -5.5 m w.e. in 81 days close to the terminus of Findelgletscher (FIN 1). Different
camera stations reveal different melt rates and total ablation, which generally depend on the station's elevation. However,
stations at different elevations can have similar melt rates as well. For example, station RHO 4 at 2589 m experienced an
average melt rate of $-0.047 \, \text{m w.e.d}^{-1}$, while the ice FIN 2 at 3015 m melted at $-0.043 \, \text{m w.e.d}^{-1}$ on average during the
common uptime (we count only days with net ablation). Further, the station on Glacier de la Plaine Morte has the lowest average
melt rate, despite not being the station at highest elevation. We assume that this might be due to meteorological conditions
such as local cold air pool formation and the Massenerhebung effect. The Massenerhebung effect describes the tendency

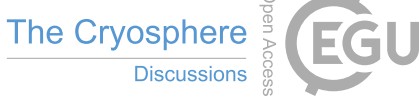

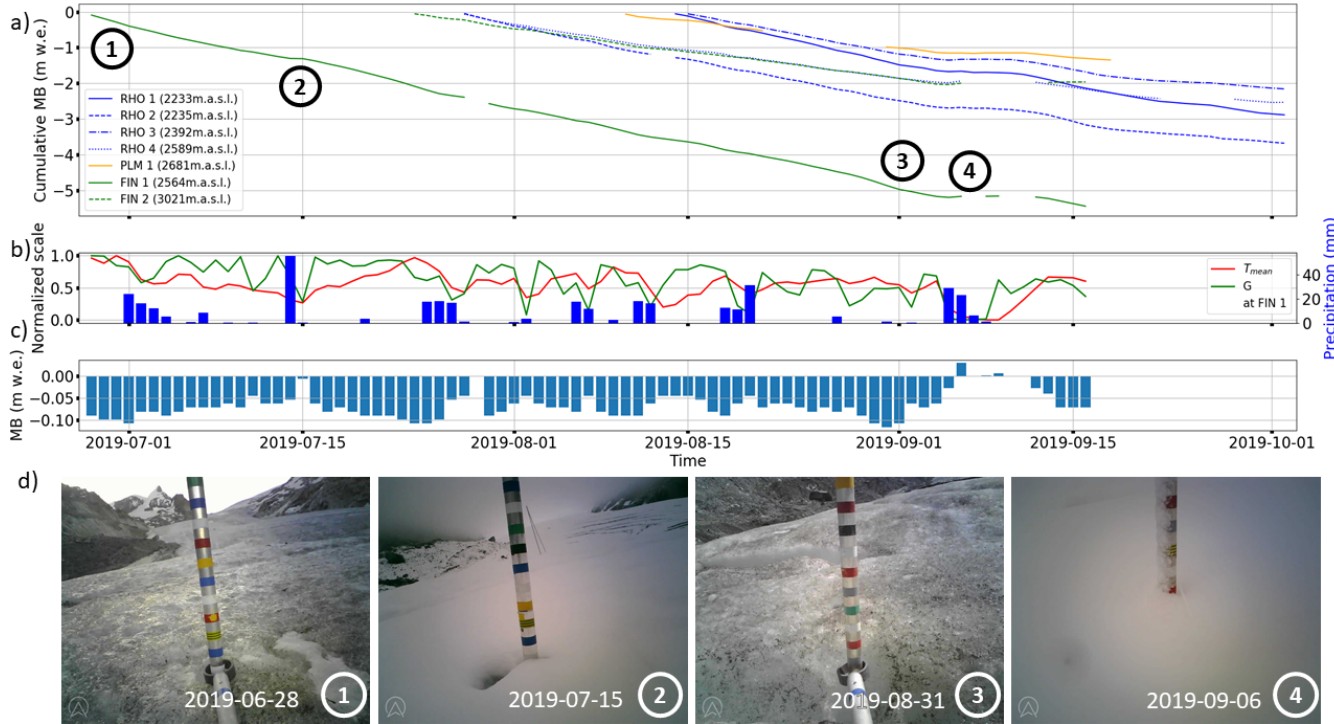

**Figure 7.** Panel a) shows the cumulative mass balance at individual camera stations during summer 2019. For comparison, panel b) shows for the longest observation series, station FIN 1, mean temperature $T_{mean}$ and shortwave radiation $G$ normalized to their respective ranges, as well as precipitation. Panel c) shows the observed daily mass balance rate at station FIN 1, and panel d) shows impressions as recorded by the camera at station FIN 1: (1) shows the camera right after setup, (2) illustrates the glacier after a light snowfall, (3) is a picture from the day with the highest melt (0.12 m w.e.), and (4) shows a stronger snowfall event hampering the stake read-out.

of higher temperatures to occur at the same elevation in the inner Alps than on their outer margins (Barry, 1992). For all
385      stations, the monthly average over the daily melt rates does not reveal big differences between July and August ($0.073 \pm 0.012$ m w.e. d$^{-1}$ vs. $0.062 \pm 0.011$ m w.e. d$^{-1}$), while daily average melt is 0.02-0.03 m w.e. d$^{-1}$ lower ($0.044 \pm 0.014$ m w.e. d$^{-1}$) in September. On Glacier de la Plaine Morte, the difference is even more pronounced with a drop of 0.06 m w.e. d$^{-1}$ in average daily melt between August and September. This is probably because, again, due to cold air pool formation and the Massenerhebung effect, Glacier de la Plaine Morte is expected to have less melt than the other station sites. On average, the

390      range between minimum and maximum melt on a particular day at all stations was 0.035 m w.e. d$^{-1}$, with values occurring from 0.005 to 0.081 m w.e. d$^{-1}$. The highest range occurred on September 1st, 2019, in connection with the passage of a convergence line/cold front (German Meteorological Service, 2019): While Glacier de la Plaine Morte was already under the influence of cooler weather, Findelgletscher and Rhonegletscher experienced another melt-intensive day. The variability at



individual stations, measured as standard deviation of a 14-day running mean, was in general low during July and August (0.016 m w.e. d$^{-1}$), while it increased at the beginning of September with the onset of intermittent snowfalls (0.026 m w.e. d$^{-1}$).

As depicted by the example of camera pictures from station FIN 1, summer 2019 is characterized by a variety of events, reaching from very hot, melt-intensive days to some fresh snow events at high elevation. The time series of normalized mean daily temperature and shortwave radiation at station FIN 1 illustrate that two heat waves have occurred at the end of June and end of July 2019. The total amount of water released by snow and ice melt on glaciers in Switzerland during these heat waves was 0.8km$^3$, which approximately equals to the annual amount of drinking water consumed in the country (Swiss Academy of Sciences, 2019). These extreme phases are also mirrored in the melt observations of our stations, as depicted in Figure 7 for FIN 1: daily melt rates peaked between 0.09 and 0.12 m w.e.d$^{-1}$ in these periods. If "heat wave" is defined as the range-normalized temperature exceeding 0.8 of the maximum temperature during the uptime of station FIN 1, the average melt rate at that station is 0.1 m w.e.d$^{-1}$ during nine days. Modelled melt across the entire glacier during these nine days based on the assimilated observations indicates the release of $6 \cdot 10^6$ m$^3$ of meltwater. Apart from the two heat waves, another phase with very high melt rates occurred at the end of August. Here, temperature and radiation are average (at 0.6 and 0.5 quantiles of their highest values), and it is unclear what exactly has caused the strong melt during this period. We first considered rain events that have not been captured by the meteorological grids, but were visible on the camera images between August 28th and August 31st, 2019. However, this assumption is speculative since neither the rain amount was a lot, nor can we prove that the rain was warm and transported a lot of energy to the glacier surface. As opposed to the extreme melt phases, there were also two interruptions by snowfalls of different strengths: from small amounts as can be seen on image 2 on Figure 7 from July 15th, 2019, to several days of intermittent snowfalls summing up to 0.25 m snow height as shown on image 4 on Figure 7.

## 4.2 Particle filter mass balance validation

Besides the direct observations presented above (Section 4.1), our framework enables us to provide predictions of daily mass balance. In this Section, these predictions are validated against reference forecasts (Section 4.2.1) and cross-validated against test-subsets of the observations (Section 4.2.2) to obtain quantitative information about their reliability. The validations are done at the camera locations. At last they are compared to glacier-wide mass balances reported by GLAMOS for the respective glaciers (Section 4.2.3).

### 4.2.1 Validation against reference forecasts

We consider two types of reference forecasts: first, we produce a forecast with the mean of annual glacier-wide melt parameters obtained from past calibration (Section 3.2), and the precipitation correction factor $c_{\mathrm{prec}}$ being constrained by the GLAMOS winter mass balance analysis of the mass budget year 2019. Second, we produce forecasts with a partially informed model, which includes the same constraint to reproduce the winter mass balance for $c_{\mathrm{prec}}$, but also a tuning of the melt parameters on one further intermediate point measurement. In our case, this intermediate measurement is the cumulative mass balance between September 2018 and 2019 at the mass balance stake closest to each camera station (locations on Figure 1). Since there are up to four stake readings per glacier, we calculate single parameter sets tuned to reproduce all possible combinations of



stake readings per glacier. This results in 19 CRPS values in total, for which we calculate the median. Further, we also consider the cases of taking meteorological input uncertainties for the reference forecasts into account and omitting the meteorological input uncertainties.

Additionally, we calculate the CRPS for these two reference forecasts by inserting two different values into the CRPS equation: (a) the mass balance of each day separately, and (b) the cumulative mass balance. For the particle filter there is no need to make this distinction, because the daily deviation from a mass balance observation also equals the deviation from the cumulative observations. Figure 8 shows the results of the validation.

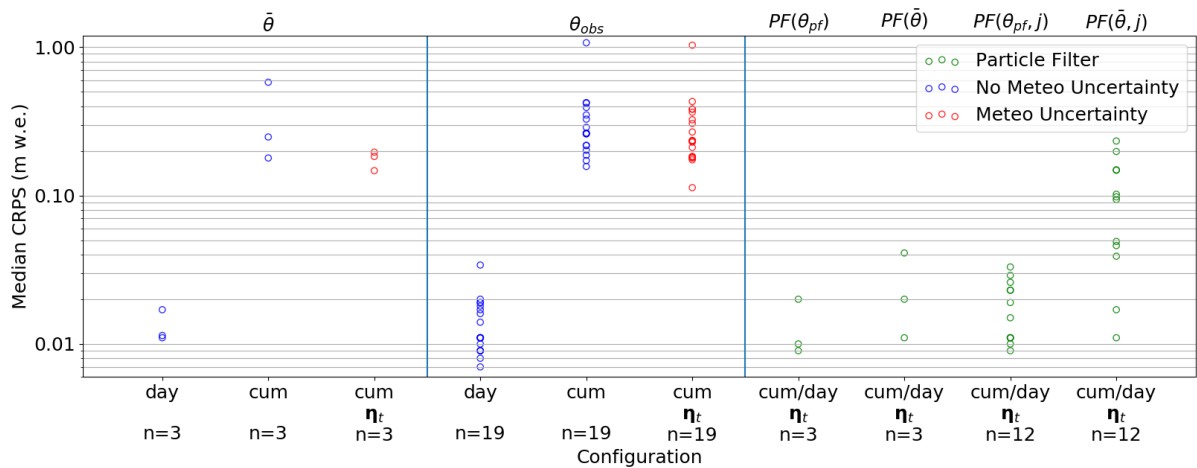

**Figure 8.** Median CRPS values over "n" validation cases for different forecasts. $\bar{\theta}$ stands for the mean parameters from past calibration, $\theta_{\mathrm{obs}}$ stands for the parameters calibrated on different combinations of mass balance stake observations close to the cameras, $\theta_{\mathrm{pf}}$ stands for the parameters found with the particle filter, blue dots stand for an assessment without respecting the uncertainty in the meteorological uncertainty, red dots and analyses indicated with "$\boldsymbol{\eta}_t$" include these sources of uncertainty, "cum" stands for the error with respect to the cumulative mass balance curve, and "day" stands for the errors in daily mass balance predictions. For clarity, the particle filter results are highlighted in green. For the particle filter, the label "cum/day" stresses that the daily prediction error equals also the cumulative prediction error, and "j" indicates cases where the particle filter was run with only one model.

    For the particle filter, daily and cumulative melt observations are in general reproduced well with an average CRPS of

0.013 [0.013] m. At the end of the assimilation period, Rhonegletscher has an average CRPS of 0.02 m, which is almost double the CRPS of the other two glaciers with slightly less than 0.01 m each. The high value of Rhonegletscher is related to the switching on of cameras RHO 1 and RHO 3, since before Rhonegletscher also has a CRPS of 0.01 m. Poor predictive performances also occur after snow has fallen, which can probably be explained with the uncertainties connected to the mass balance stake readings during these times. We have run experiments where the particle filter is limited to using mean parameters

and/or single models instead of parameter distributions and the full model ensemble. In more than half of the experiments, the resulting CRPS values are higher than the highest CRPS obtained with the full setting. The experiments thus show that it is beneficial to include all four models and parameter uncertainty into the filter.



Comparing the CRPS of the particle filter with the reference forecasts, the performance closest to the particle filter is delivered by the daily mass balance forecast produced with mean melt parameters and no uncertainty in the meteorological

input (mean CRPS = 0.013 [0.013] m). However, as soon as the CRPS is calculated from the cumulative mass balance produced with mean melt parameters, the CRPS increases to 0.335 [0.241] m on average. This is due to the fact that with the mean parameters the prediction is not able to adapt parameters to meteorological conditions over time. Like this, the cumulative mass balance can temporarily be under- and overestimated or diverge completely from the cumulative observations. Somewhat counterintuitively, but for the same reason, the CRPS is on the same order when parameters have been tuned to match the

nearby stake readings. For the cumulative deviation, we find CRPS values of 0.297 [0.298] m w.e. with considering and 0.321 [0.316] m w.e. without considering meteorological uncertainty, respectively. The CRPS of daily mass balances produced without considering meteorological input uncertainty is roughly the same compared to both the particle filter prediction and the prediction with mean melt parameters (median CRPS: 0.012 [0.014] m w.e.).

The particle filter improves the performance scores of the reference forecasts by 91% to 97% for the individual glaciers, with

the exception of daily forecasts, where it performs roughly equal. Most importantly though, with the particle filter it is possible to give daily uncertainty estimates during the assimilation process without further calculations, which is a clear advantage of the particle filter over the methods that do not account for uncertainties. Especially for the application of our framework the quantification of melt uncertainty is essential.

### 4.2.2 Cross-validation

As opposed to validating the particle filter against reference forecasts, it is also possible to run the particle filter with only subsets of the observations as input and evaluate mass balances at test locations on the same glacier. In our case, we split the existing observations on a glacier into subsets by station, where a test subset always contains the observations from one station (leave-one-out cross-validation). Figure 9 shows the CRPS over time when predicted at the test locations.

We perform this kind of cross-validation and find that in general the cumulative mass balance at the test locations follows

the cumulative observations curve well, but not as closely as when the test location's data assimilated with the particle filter. For Findelgletscher we find 8.4% average deviation (median CRPS 0.046 and 0.206 m w.e.) when comparing the cumulative mass balance curve with the particle filter prediction, while for Rhonegletscher the average deviation at the test locations is 6.5% average deviation (median CRPS 0.091, 0.010, 0.198 and 0.165 m w.e.). The highest CRPS values for Rhone stem again from the period after mid of August, when two additional cameras have been set up. However, these values are on average still

better than the reference forecasts in Section 4.2.1.

These results show the ability of the particle filter to reproduce observations also at locations on the glacier from which it has not received any input in the form of observations. However, the performance is not as good as when trained with all observations. It also becomes obvious that even with an augmented particle filter, which is able to adapt parameters over time and with the input of observations from different locations, it is demanding to find a unique, glacier-wide parameter set that

can reproduce mass balances equally well at all locations on the glacier.





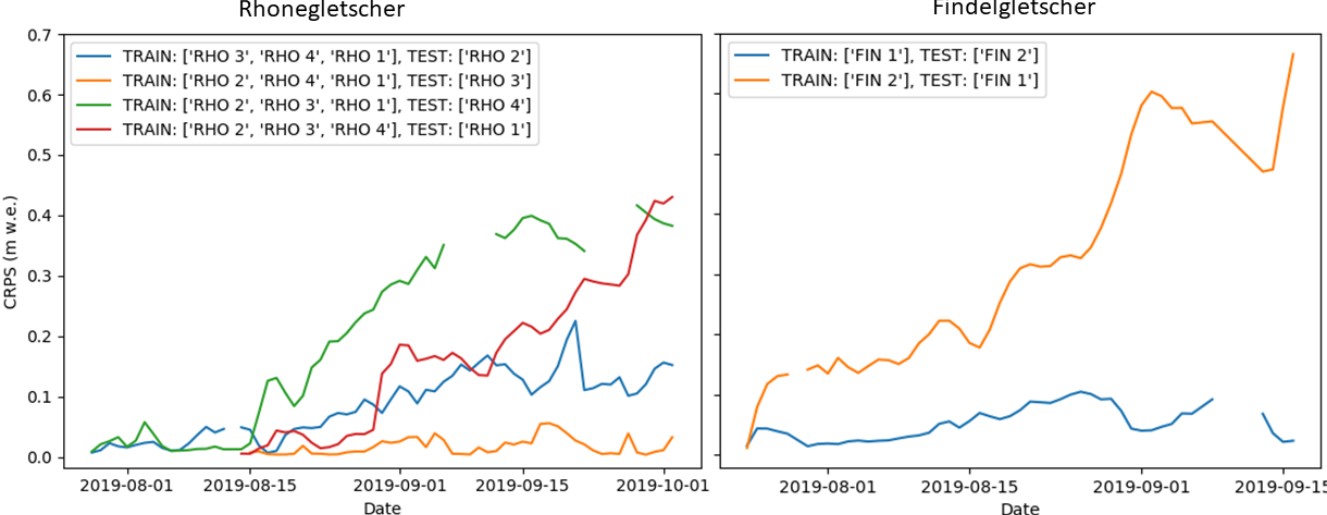

**Figure 9.** The CRPS values over time when predicted in a leave-one-out cross-validation procedure on Rhonegletscher and Findelgletscher.

### 4.2.3 Comparison to GLAMOS glacier-wide mass balances

We compare assimilated model ensemble predictions to the glacier-wide annual mass balance reported by GLAMOS at the autumn field date of the mass budget year 2019. It is therefore necessary to couple the particle filter period with a free model run period that begins at the field campaign date in autumn 2018. Figure 10 illustrates these periods with different model and
parameter settings.

During the free model run period, we calculate mass balance only with the parameters that were calibrated in the past (Section 3.2), which results in about 45 distinct model runs. We use this first period to provide initial conditions for the particle filter period, which lasts from the first camera setup on a respective glacier until cameras are retrieved, or the autumn field date, respectively. To achieve a random coupling of the initial conditions with the initial particles during the particle filter period,
we sample 10000 times from the initial conditions at the first camera setup date. However, not all free model runs have to be used: they can also be pre-selected based on the cumulative mass balance observations that have been measured at the mass balance stakes close to the camera stations. For this case, we select model runs that reproduce these observations at the stake elevation within an estimated reading uncertainty of $\pm 0.05$ m w.e.. By combining the free model run period with the particle filter period for these two cases, we calculate the cumulative mass balance between the autumn field date 2018 and the autumn
field date 2019, which are compared to the GLAMOS analyses in Table 3.

It is worth noting that for the assimilated estimates, 83-95% (83-96%) of the total uncertainty stem from the period before the particle filter was initiated. For the particle filter mass balances without pre-selection of initial conditions, the agreement with the GLAMOS analyses varies between a difference of 0.41 m w.e. for Findelgletscher and a good agreement for Plaine Morte and Rhonegletscher. For the case with a pre-selection the absolute difference to the GLAMOS values even changes by



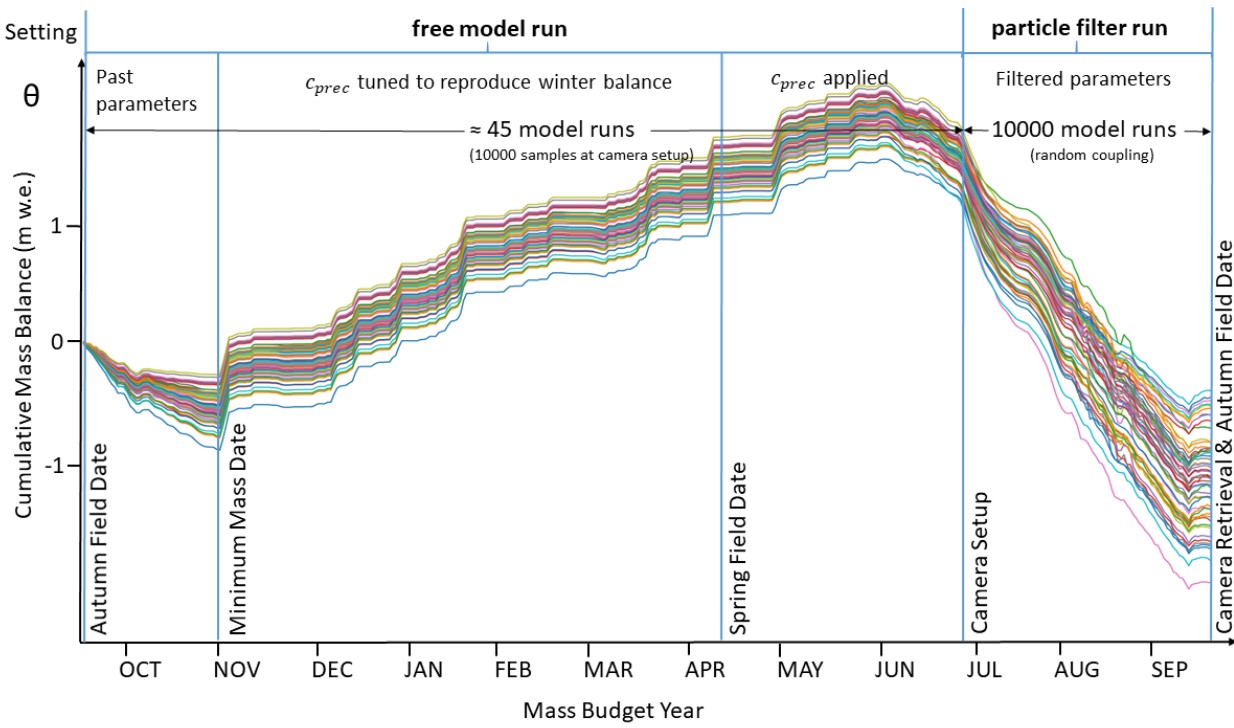

**Figure 10.** Schematic model and parameter settings on Rhonegletscher during the mass budget year 2019. After an initial phase with parameters from past calibration, the precipitation correction factor $c_{prec}$ is tuned to match the winter mass balance. When the first camera is set up, we sample the existing model runs 10000 times to be able to couple the free model runs with the 10000 particles during the particle filter period (not all are drawn for readability).

**Table 3.** Mass balances calculated between the autumn field date 2018 and the autumn field date 2019 for the particle filter and the values reported by GLAMOS. Uncertainty values are given as standard deviations.

| Glacier | Particle filter (no pre-selection) (m w.e.) | Particle filter (pre-selection) (m w.e.) | GLAMOS (m w.e.) |
|---------|---------------------------------------------|------------------------------------------|-----------------|
| PLM | $-1.99 \pm 0.46$ | $-1.89 \pm 0.17$ | $-1.77 \pm 0.09$ |
| FIN | $-0.65 \pm 0.14$ | $-0.46 \pm 0.30$ | $-0.24 \pm 0.16$ |
| RHO | $-0.68 \pm 0.30$ | $-1.07 \pm 0.28$ | $-0.77 \pm 0.20$ |

$-0.10$, $-0.19$ and $+0.39$ m w.e., respectively, although the sign of the difference can change. Consequently, including the stake mass balance readings can also have a negative effect on the agreement with the GLAMOS analyses. A reason for this can be either that the mass balance stakes are not at the observation locations, but up to 500 m away from the camera stations (as in the case of RHO 3), or that the mass balance gradients of the pre-selected runs are unfavorable. Overall differences to the GLAMOS analyses can be explained by (1) the difference in the individual approaches to calculate glacier-wide mass balance



from point observations, (2) the use of only 1-4 point observations biased to the ablation zone to compute glacier-wide mass balance in this study versus a complete network of 5-14 stakes over the entire elevation range used in the GLAMOS analyses, (3) lack of representativeness of the camera observations for the accumulation zone of the glaciers, (4) lack of representation of individual winter accumulation measurements in our glacier model, or (5) a problem with representing the mass balance of the glacier with only one parameter set.

## 4.3   Individual model performance

We analyse model performance by looking at model probabilities $\pi_{t,j}$ and model particle numbers $N_{t,j}$ of the four melt models over time at individual glaciers. High model performance is indicated by high probabilities and particles numbers over long time periods.

Figure 11 shows the model performance of all four melt models at all three glacier sites. In general, it occurs for all models

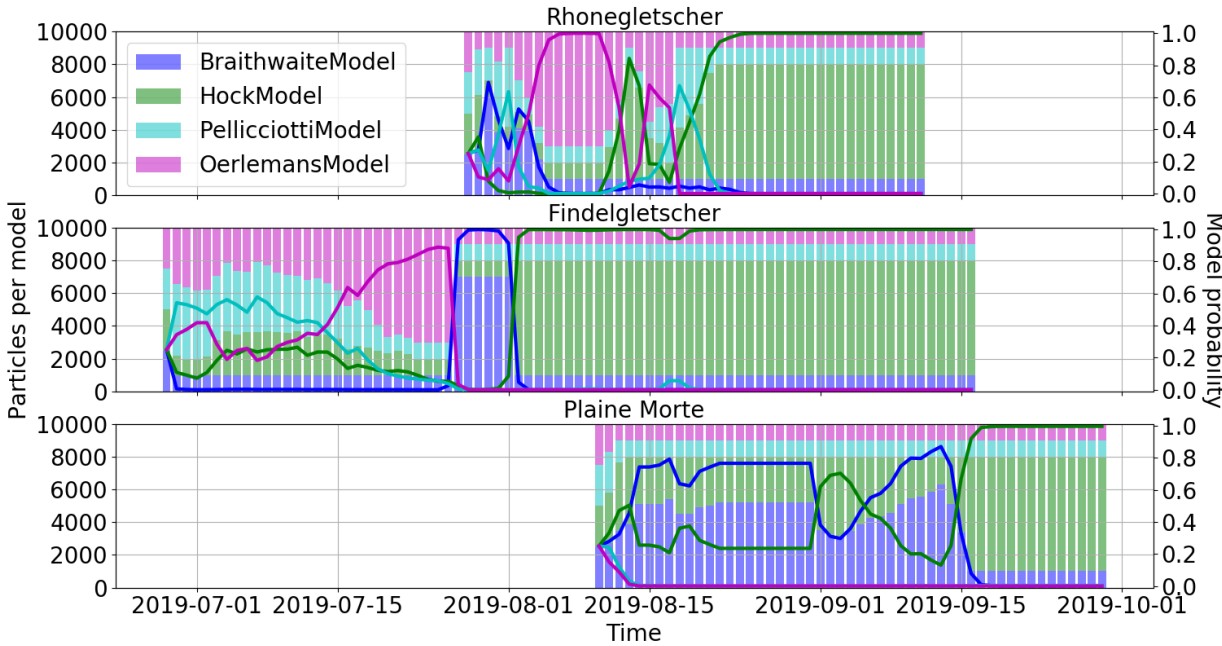

**Figure 11.** Model probabilities (lines) and model particles (stacked bars) for the three modelled glaciers over time.

and glaciers that a model is not removed from the ensemble in the resampling step, when the model performs poorly. It also occurs several times that models recover and show good performances at a later stage again, most prominently for example the HockModel on Rhonegletscher and Findelgletscher. Since this evolution from poorly performing to recovered, well performing model occurs, the resampling procedure introduced in Section 3.3.5 proves to be useful.





During most of the times there is one or two models that dominate the ensemble prediction, where we define "model dominance" as a model probability greater than 0.5. Averaged over all glacier and time steps, the HockModel has the highest model probability (0.58), while the BraithwaiteModel has an average model probability of 0.19, the OerlemansModel of 0.15 and the PellicciottiModel of 0.08. The fact that the BraithwaiteModel has the second highest average probability - even though OerlemansModel being close - can possibly point to the fact that there currently, the calculation of the albedo might not be accurate enough, such that the BraithwaiteModel, which does not use albedo as an input, can profit from this potential inaccuracy. The reasons why the HockModel has higher probabilities than the two models that use the actual incoming surface radiation may be manifold, but here we speculate that this may be linked to two circumstances: first, it might happen that the HockModel has by calibration a broad enough prior parameter distribution, which allows it to be the best performing model for all occurring combinations of meteorological input and observed melt. Second, it might be that $I_{\mathrm{pot}}$ is less error-prone than $G$ due to the fact that it is not subject to potential processing uncertainties, e.g. through cloud masking. Although it is not a real meteorological forcing, potential irradiation can be computed on a grid with high resolution. As opposed to that, the shortwave incoming solar radiation from MeteoSwiss is derived from satellite data with a coarser kilometer-resolution.

In terms of the temporal evolution, for Rhonegletscher and Glacier de la Plaine Morte the model dominance is determined already within the first few days and stabilizes then. However, model dominance can obviously also swap easily though, meaning that within a short time period of three days or less another model becomes dominant. This can be observed for all glaciers at different points in time. For example, model dominance swaps on all glaciers to the HockModel on different days, while for Rhonegletscher the HockModel even had a model probability close to zero before. With the given data, it cannot be answered why the HockModel then stays dominant throughout September for all glaciers. There is also a clear indication that setting up a new camera might have an influence on model probabilities (July 24th on Findelgletscher, August 13th on Rhonegletscher). Surprisingly little influence on the model dominance was exerted by snowfall events (e.g from September 9th to 17th on Findelgletscher, or from September 5th to September 11th on Rhonegletscher).

## 5 Conclusions

In this study, we have mounted seven cameras on three Swiss glaciers, delivering 352 point mass balance observations throughout the summer 2019. At the camera locations, we have observed melt rates up to $0.12\,\mathrm{m\,w.e.d}^{-1}$ and up to more than $5\,\mathrm{m\,w.e.}$ total melt in 81 days. To calculate near real-time mass balances for the equipped glaciers, we made use of mass balances ensemble modeling with three temperature index models, a simplified energy balance model and uncertain model inputs for all models. Additionally, we used a particle filter scheme to assimilate the camera observations into the model ensemble. The particular focus was put on delivering a stable ensemble, that can be applied to monitor mass balance throughout the summer. To obtain these results, it was necessary to make considerations about model parameter variability as well as the particle filter stability. For the former, we use a distribution of parameters from past model calibration as prior input to an augmented particle filter, which is also able to estimate parameters while assimilating observations. For the latter, we designed the particle filter such that temporarily poorly performing models in the ensemble can recover at a later stage. At the end of the mass budget year





2019, we find for Glacier de la Plaine Morte $-1.89$ m w.e., for Findelgletscher $-0.46$ m w.e., and for Rhonegletscher $-1.07$ m w.e..

We have found that the mass balances given by the particle filter are about as close to the actual cumulative observations

(Continuous Ranked Probability Score= 0.013 m w.e.) than for two reference forecasts, where either no measurements are available or only one intermediate set of stake readings have been made. However, the particle filter improves the performance scores of reference forecasts by 91% to 97% when considering cumulative mass balance observations. Moreover, the particle filter is able to deliver direct uncertainty estimates. These can help, e.g., to better assess uncertainties in runoff if the mass balance is used as input to hydrological models. In a leave-one-out cross-validation procedure on the individual glaciers we

showed that the particle filter does not deviate more than 8% from the cumulative mass balance observations at the test locations. In an analysis of the individual model performance, we found that our technique to prevent models from being removed from ensemble is useful, since models can recover at a later stage. The temperature index model by Hock (1999) has the highest model probability on average (0.58), while the ensemble model probabilities can also swap suddenly on particular days. We assume that for example the setup of a new camera can be responsible for such a swap in model probabilities.

We aim for an extension of the particle filter scheme in a next step, where we constrain glacier mass balances and model parameters by using remotely sensed observations of albedo and snow lines. These measurements are indirect, but have the potential to (1) complement the camera measurements extensively and to (2) overcome the limited knowledge about the spatial and temporal extrapolation of glacier mass balances and model parameters.

*Code and data availability.* The camera observations used can be obtained from the authors upon request, the meteorological data can

be obtained from https://www.meteoschweiz.admin.ch, and the glacier outlines and mass balances are available free of charge from https://www.glamos.ch/. The code used to produce results and figures will be published at a later stage within the CRAMPON framework and can, until then, be obtained from the authors upon request.

*Video supplement.* Time lapse videos of all camera observations used in this study are available as videos under the following DOIs: PLM-1: https://doi.org/10.5446/48826, FIN-1: https://doi.org/10.5446/48824, FIN-2: https://doi.org/10.5446/48825,

RHO-1: https://doi.org/10.5446/48820, RHO-2: https://doi.org/10.5446/48821, RHO-3: https://doi.org/10.5446/48822, RHO-4: https://doi.org/10.5446/48823

## Appendix A: Handling of multiple cameras

Assume that camera $i$ is installed at elevation $z_i$ on day $t_{i-1}$ where $t_0 < t_1 < t_2 \ldots$ (to be coherent with earlier notation that the first camera is installed at time $t_0$). From time $t_{i-1}$ onwards, we include $b_{\mathrm{sfc}}(t_{i-1}, z_i)$ in the state vector as a component which





remains constant. Then the observations at time $t > t_{i-1}$ are functions of the state at time $t$:

$$h(t, z_i) = \frac{b_{\mathrm{sfc}}(t, z_i) - b_{\mathrm{sfc}}(t_{i-1}, z_i)}{\rho_{\mathrm{ice}}} + \epsilon(t, z_i). \tag{A1}$$

The true value of $b_{\mathrm{sfc}}(t_{i-1}, z_i)$ is unknown, and the uncertainty is represented by the values $b_{\mathrm{sfc},k}(t_{i-1}, z_i)$ of the particles. Thus at time $t$, the contribution from the observation $h(t, z_i)$ to the weight of particle $k$ is proportional to

$$\exp\left(-\frac{(h(t, z_i) - (b_{\mathrm{sfc},k}(t, z_i) - b_{\mathrm{sfc},k}(t_{i-1}, z_i))/\rho_{\mathrm{ice}} \cdot \rho_{\mathrm{w}})^2}{2\sigma_\epsilon^2}\right). \tag{A2}$$

Although $b_{\mathrm{sfc},k}(t_{i-1}, z_i)$ never changes during the propagation step, it will change in the resampling steps. Thus the uncertainty about $b_{\mathrm{sfc}}(t_{i-1}, z_i)$ will decrease as time proceeds. This is presumably not realistic, but the effect of small errors in the baseline also diminishes as time proceeds.

## Appendix B: Resampling procedure

The technical details of the resampling procedure in Section 3.3.5 are the following: if, after prediction and update, $N_{t,j}$ denotes
the number of particles with model index $j$, we prevent models from not being resampled by choosing a minimum model contribution $\phi < \frac{1}{4}$ to the ensemble. This ensures that the resampling step preserves a minimum particle number $N_{t,j} \geq \phi N_{\mathrm{tot}}$ representing model $j$. For our application, we choose $\phi = 0.1$. If the posterior probability of model $j$ (Equation 18) is smaller than the minimum contribution $\phi$, an unweighted sample that represents $\pi_{t,j}$ correctly, must have less than $\phi N_{\mathrm{tot}}$ particles with model index $j$. To ensure our minimum contribution condition though, we generate a weighted sample $(\tilde{\boldsymbol{x}}_{t,k}, \tilde{w}_{t,k})$,
such that each model index $j$ appears at least $\phi N_{\mathrm{tot}}$ times and the weights are as close to uniform as possible. We select the particles $\tilde{\boldsymbol{x}}_{t,k}$ in a two step resampling procedure: first, the number $N_{t,j}$ of particles with model index $j$ is chosen to be $N_{t,j} = \phi N_{\mathrm{tot}} + L_{t,j}$, where $L_{t,j}$ are excess frequencies. We obtain these frequencies by sampling a total of $N_{\mathrm{tot}}(1 - 4\phi)$ model indices from $\{1, 2, 3, 4\}$ with weights proportional to how much a model probability exceeds the chosen minimum contribution, i.e. $\max(0, \pi_{t,j} - \phi)$. In a second step, we draw for each model a resample of size $N_{t,j}$ with weights $w_{t,k}/\pi_{t,j}$ from the particles
with model index $j$. The combined set of the $N_{\mathrm{tot}}$ resampled particles gives the new filter particles $\tilde{\boldsymbol{x}}_{t,k}$.

However, introducing a restriction on the minimum number of particles per model can lead to biased estimates, as poor models with probability $\pi_{t,j} \leq \phi$ are overrepresented in the ensemble. To compensate that poor models occur too often among the resampled particles (and the other models not often enough), the following weight has to be given to $\tilde{\boldsymbol{x}}_{t,k}$:

$$\tilde{w}_{t,k} = \frac{\pi_{t,j}}{N_{t,j}} \text{ if } \tilde{m}_{t,k} = j. \tag{B1}$$

These weights sum to unity and preserve the original weights $w_{t,k}$ on average. Since they can become very small though, we work with the logarithm of the weights to avoid numerical underflow. It should be noted that we insert $\tilde{w}_{t-1,k}$ for $w_{t-1,k}$ in Equations (11) and (17). In order to see that the weights we choose for $\tilde{\boldsymbol{x}}_{t,k}$ are correct, denote the number of times the particle $\boldsymbol{x}_{t,k}$ is selected in the resampling procedure by $\tilde{M}_{t,k}$. This means that the resampling gives $\boldsymbol{x}_{t,k}$ the random weight $\frac{\tilde{M}_{t,k}}{N_{\mathrm{tot}}}$,



which is then multiplied by the additional weight $\tilde{w}_{t,k}$. Hence $\boldsymbol{x}_{t,k}$ receives the total weight

$$w'_{t,k} = \tilde{w}_{t,k} \frac{\tilde{M}_{t,k}}{N_{\text{tot}}}. \tag{B2}$$

If $m_{t,k} = j$ it holds that

$$E(w'_{t,k} \mid N_{t,j}) = \tilde{w}_{t,k} E(\tilde{M}_{t,k}/N_{\text{tot}} \mid N_{t,j}) = \frac{\pi_{t,j}}{N_{t,j}} \frac{w_{t,k} N_{t,j}}{\pi_{t,j}} = w_{t,k}, \tag{B3}$$

i.e. on average the new weights $w'_{t,k}$ are equal to the original weights.

*Author contributions.* JL had the particle filter idea, implemented all models, did all figures and wrote the paper. HK supervised the particle
filter methodology, brought in the method to prevent models from disappearing from the ensemble, and reviewed the paper. MH commented
on the method, reviewed the paper and mounted some of the stations. CO prepared and mounted most of the stations. MK commented on the
particle filter and reviewed the paper. DF did the overall supervision, proposed to use data assimilation in JL's doctorate, commented on the
method, reviewed the paper and acquired the funding.

*Competing interests.* The authors declare that they have no conflict of interest.

*Acknowledgements.* We would like to acknowledge the funding that we got from Global Climate Observing System (GCOS) Switzerland and
the extensive support that we got from the manufacturer of the cameras and transmitter boxes, Holfuy Ltd (in particular Gergely Mátyus).
We would like to thank the teachers of the Joint ECMWF and University of Reading Data Assimilation Training Course that helped to
significantly improve JL's knowledge on data assimilation, in particular Javier Amezcua. Further, we would like to thank Anastasia Sycheva
and Emmy Stigter for test-reading the methods chapter for reader friendliness. We appreciate the help from all people that conducted the
field work apart from the authors, namely Małgorzata Chmiel, Amaury Dehecq, Lea Geibel, Katja Henz, Serafine Kattus, Johanna Klahold,
Claudia Kurzböck, Amandine Sergeant and Michaela Wenner, and we thank all people that took part in the round robin experiment apart
from the authors, namely Amaury Dehecq, Eef van Dongen, Elias Hodel, Jane Walden and Michaela Wenner. We would also like to thank
Bertrand Cluzet and colleagues for having changed the acronym of their project which coincidentally was the same as ours (CRAMPON).





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
