# Peer review of "Assimilating near real-time mass balance stake readings into a model ensemble using a particle filter"

_The Cryosphere, 2020_

## Referee Comment (RC1) · Douglas Brinkerhoff (Referee) · 8 Feb 2021

**Summary**

In 'Assimilating near real-time mass balance observations into a model ensemble using a particle filter,' Landmann and co-authors describe the installation of a set of cameras aimed at measuring point ablation rates at several locations in the Alps, and then assimilate those measurements into an ensemble of ablation models using a novel implementation of a particle filter. They compare their model results favorably with a much more laborious empirical mass balance measurement for each of their three glaciers.

[Figure]

The key ideas of this paper are excellent and important. First, the use of telemetered cameras to provide continuous measurements of melthas the potential to substantially improve the temporal and spatial resolution of monitoring in regions where it is feasible. Second, the probabilistic assimilation of these observations into models is a clear advancement in the way that data is extrapolated into broader conclusions.

From a scientific perspective, I think that the paper is sound. There are statistical modelling choices that I disagree with and that I hope that the authors will address, but this can be done through added discussion in the text rather than any new analysis or methodology. From a stylistic perspective, I hope that the authors will carefully look through the paper and critically identify jargon and unclear descriptions; the paper would make a more enthusiastic reader if the language were simplified as much as possible. I have made specific comments in relation to both of these points below.

**Comments**

**Title** The observations are not of mass balance, but of surface elevation (specifically in the negative direction). I suggest changing the title to be more precise.

**L10** The reader does not yet know what 'model probability' is in the abstract, nor is the abstract notion of 'custom resampling' useful here.

**L39** Of the three points (first, second, third) made after this line, only one logically follows this statement.

**L49** List of references should have an e.g. in front of it. There are many other examples of ensemble modelling for ice sheet projection.

**L55** 'discussed how' → 'not clear whether'(?)

**L63** surface point mass balance → surface point ablation. You don't measure mass balance, you measure volume change in one direction.

**L80** as above.

**L103** 'cumulative surface height change' is (mathematically) equivalent to 'surface height'. I suggest the latter for brevity.

**Eq. 1** This equation is only valid for bare ice. This is briefly touched on elsewhere, but should be reiterated here. In fact, it might be better to state that the operation operation relates $h(t, z)$ to $a_{sfc}$.

**L111** 'Short snow events ...'. We never see this notion of assigning a high uncertainty SWE estimate again. Is this actually done, and specificall how?

**L133** I'm confused by the lapse rate thing. Why don't you continue to be a Bayesian and just use the probability distribution over the lapse rate inferred from the data without injecting questionable notions of 'significance'? This could then be propagated into downstream analysis.

**L151** In what sense is an outline a surface? I don't understand this line.

**L158** 'Values of glacier-wide mass balance ...' I don't understand what 'partly harmonized' means in this context?

**Eq. 2** Perhaps it's standard notation, but having $c_{prec}$ mean an entirely different thing (with different units) than $c_{cfc}$ is really confusing.

**L214** It would be useful to make a note that $G$ is a function of $I$.

**Eq. 8** Suggest using $\Delta t$ rather than $dt$, as the latter is usually reserved for infinitesimals.

**Eq. 10** The 'general framework' also has $\epsilon$ inside of $\mathcal{H}(x)$, although it doesn't appear that way in this work.

**Sec. 3.3** I find it confusing that the parameter update process appears at the end, even though Figure 6 indicates that it happens at the same time as the state prediction.

**L289** It might be clearer to state explicitly that a particle is always associated with only 1 model over its "lifetime".

**Eq. 16** I strongly disagree with the choice of setting $\beta_t = 0$. This is because this is tantamount to the assumption that the model predictions are perfect, which is certainly not the case. In reality, two models are only different in their reliability to the extent that their predictions differ by more than their internal uncertainty and one fits the data better than the other. Setting this model error to zero artificially accentuates the differences in the likelihoods computed for different model and encourages the mode collapse (what you call 'model dominance') exhibited in Figure 11.

**Table 2 Caption** By covariance, do you mean standard deviation?

**L320** the standard symbol for variance would be $\sigma_\epsilon^2$.

**Eq. 18** An implicit assumption made throughout is that a single model's probability is marginally uniform, or alternatively that $P(m_t) = \mathrm{Dirichlet}(\mathbf{1})$, to wit that one model being dominant is just as probable as all four models contributing equally. This is a weird assumption for a time dependent problem, because it means that physical reality is subject to sudden switches between governing principles. Again, this leads to the mode collapse seen in Fig. 11. Predictions might be made substantially more robust by putting a prior on $P(m_t)$ such that the more probable case is an averaging of the four models, and deviation from that has to be the result of significant evidence.

**L338** It's not that there's no stochasticity, it's that $m_t$ for a given particle doesn't evolve at all!

**Sec. 3.4** This section is essentially incomprehensible, with the section on proper scoring reading like it was pasted from a statistical methods paper. This being the Cryosphere, it's important to try to help your reader with some intuition as to what the CRPS actually means, and why its potential impropriety matters. A figure describing the metric might be useful, or perhaps a simple example describing circumstances where the value is high or low. While the rest of the paper is still accessible not understanding CRPS, the analysis breaks down to 'big number bad, low number good,' which is unfortunate given that there is probably much more insight to be gained from the following sections.

**Sec. 4.2.1** This section is quite unclear, specifically what the differences are that these include relative to the 'full' forecast.

**L435** Perhaps I missed it, but I can't find anything describing what the number in brackets means.

**Sec. 4.2.2** This section on cross-validation is very clear and good. Maybe it would be useful to comment on the temporal pattern evident in Figure 9, with CRPS increasing through time, but at different rates between different cross-validation folds.

**L481** I don't understand where the '45 distinct model runs' come from. Also, what is a 'random coupling'?

**L495–496** I don't understand this sentence, nor why conditioning initial conditions on observations leads to poorer results.

**Figure 11** To emphasize earlier comments again, this pattern of mode collapse is strongly indicative of an over-confident likelihood operating in an M-open framework. It's well known that Bayesian inference only 'works' when the models are correctly specified. For Bayesian model averaging (which is what the particle filter is doing in a time dependent way), this still holds: because the true physics are not contained in the set of equations that the filter has available to pick from, yet this additional uncertainty is not explicitly specified, the filter hops between the model that fits the observations in the moment. While I don't expect any additional analysis, I think it would be appropriate to make this assumption explicit in the text, and to perhaps reference it when describing the fast switching between dominant models.

**Figure ??** Two things that are missing from the paper are time series' of state and parameter distributions. It would be very interesting to see the evolution of uncertainty in the predictions away from observations, and also to see how quickly parameters change or revert to the mean.

---

## Referee Comment (RC2) · Anonymous Referee #2 · 17 May 2021

This manuscript describes the assimilation of surface elevation measurements into an ensemble of ablation models. The particular data assimilation (DA) technique used is a particle filter.

I think the manuscript is quite interesting, well written, and can become an important contribution to the community. On one hand, it tackles a very important and tangible issue (studying the loss of glacier mass). On the other, it uses an advanced DA method. I have some comments about the manuscript. I will be happy to recommend

this manuscript for publication once the comments are addressed.

– Major comments and questions – - What made you choose a particle filter as DA method as opposed to more traditional methods (e.g. variational methods and ensemble Kalman filter). It would complement the work if you discussed why a PF suits this problem. - Line 105. If I understood well, the observations are of a cumulative quantity. In this case, do observation errors need to consider time auto-correlations? - Line 255. I was a bit confused on where the uncertainties of the input variables are represented. Are they represented in the model error beta, in the observation error epsilon (as mentioned in line 258), or both? - The use of the PF in a multi-model ensemble context is quite interesting, especially since each model has different parameters one is trying to estimate. Is there previous work in this regard? Could you provide some references? - When discussing the particle filter, you introduce the concept of 'minimum contribution' for some particles. This is taken into account when weighting, as explained in appendix 2. There is a comment saying that the original weights are preserved 'in average'. Could you elaborate more on this statement? - Equation 21. How are $\mu_0$ and $\Sigma_0$ chosen?

– Minor comments and questions – - The title mentions 'mass balance observations', whereas the observations are of surface elevation. - Line 96. It is mentioned that the camera images are read 'manually' to obtain the daily cumulative surface height change. Is it literally reading the marks from the ablation pole? How could this be automated to be applied to more places? - Line 127. Can you say more about the 0.2 degree resolution? How does this compare with other products? Is it high or low resolution? - Figure 7. I think making the vertical axis larger for panels a and b could make the figure easier to read. - Figure 8. The individual circles are difficult to see. Please make the circunferences thicker, and maybe increase the size of the figure. - In pages 22-24 (approximately) there are several places where a quantity is written followed by () or []. It was not clear to me what the quantities in the parenthesis are, and why there are two styles.

– Typos and corrections – - Line 109. ... because it can happen that the camera construction sinks... -> ... because the camera construction can sink. (easier to read). - Line 118. melt during night -> nighttime melting - Line 190 on. When mentioning the models in an itemised list, start the sentences with capital letter. - Figure 5. Some of the words in the labels are split into two lines - In the title of table 2 it should say 'standard deviations' instead of 'covariances'.

---

## Author Comment (AC1) · 19 Jun 2021

**Author response to the review of D. Brinkerhoff**

Johannes Landmann and co-authors

June 2021

Dear Douglas Brinkerhoff,
Dear Editor,
We thank Douglas Brinkerhoff very much for the valuable comments regarding our manuscript. We really appreciate the thoughtful input and the detailed recommendations to improve the document. We address the raised requests in the form of point-by-point responses and make suggestions for how to update the final manuscript.
Best regards,
Johannes Landmann and co-authors

**RC:** *Title The observations are not of mass balance, but of surface elevation (specifically in the negative direction). I suggest changing the title to be more precise.*
**AR:** Thanks for this comment. We agree that the observations we make are in "surface elevation space" and that these are then transformed into a mass change using the density of the medium (snow/ice). However, the reviewer will agree that (i) the surface elevation change of a glacier at a given point is the result of both surface mass balance and a component due to ice emergence (see Cogley et al., 2011, p. 38), and (ii) the changes that we measure at our stakes are only due to the former. Stated differently: what we see is the surface elevation change due to glacier mass balance, but not the total ice thickness change. To avoid possible confusion, we propose to change the title into: "Assimilating near real-time mass balance stake readings into a model ensemble using a particle filter".

**RC:** *L10 The reader does not yet know what 'model probability' is in the abstract, nor is the abstract notion of 'custom resampling' useful here.*
**AR:** We understand the point and suggest the following, simplified text: "These observations are assimilated into an ensemble of three temperature index (TI) and one simplified energy-balance mass balance models using a particle filter. By using state augmentation, we assign temporally-varying weights to individual models. We analyse model performance over the observation period, and find that the model probability is highest for [...]".

**RC:** *L39 Of the three points (first, second, third) made after this line, only one logically follows this statement.*
**AR:** We agree that the logics might not have been apparent because the enumerations were stretched very far out into the subsequent text. We propose to restructure the text as follows: "In many cases, mass balance analyses are available twice a year, i.e. they are based on seasonal in situ observations (Cogley et al., 2011). This relatively low mass balance analysis frequency is mainly due to the fact that in situ observations are not often made, because they are expensive in terms of both time and manpower. Only recently have low-cost and high-frequency monitoring approaches emerged (Hulth, 2010, Fausto et al., 2012, Keeler and Brugger, 2012, Biron and Rabatel, 2019, Carturan et al., 2019, Gugerli et al., 2019, Netto and Arigony-Neto, 2019). However, even if higher observation frequencies are available with these new approaches, it is not straightforward to calculate analyses at higher frequencies. This is because near real-time estimates are often based on ensemble modelling, in order to enable a correct quantification of uncertainties. Ensemble modelling is used in glaciology in the context of model intercomparison projects (Hock et al., 2019), future projections for ice sheets and mountain glaciers (e.g. Ritz et al., 2015, Shannon et al., 2019, Golledge, 2020, Marzeion et al., 2020, Seroussi et al., 2020), and also to determine the initial conditions for modelling (Eis et al., 2019). However, ensembles are currently not prominent in the calculation of seasonal or daily glacier mass balances. Another reason why calculating higher-frequent glacier mass balance analyses is not straightforward is that there is often a lack of knowledge about the exact short-term parameters in mass balance models. This poses a problem, since e.g. temperature

index (TI) models are parametrizations of the full energy balance equation and deliver inaccurate results when applied with inapt parameters for a specific location [...]".

**RC:** *L49 List of references should have an e.g. in front of it. There are many other examples of ensemble modelling for ice sheet projection.*
**AR:** Thanks! We will insert "e.g.".

**RC:** *L55 'discussed how'→'not clear whether'(?)*
**AR:** We will exchange the wording as proposed.

**RC:** *L63 surface point mass balance→surface point ablation. You don't measure massbalance, you measure volume change in one direction.*
**AR:** Besides the answer given in reply to the comment related to the title (see our first answer), note that we do not only measure ablation but also accumulation. Together, this is the mass balance of the surface.

**RC:** *L80 as above.*
**AR:** Idem.

**RC:** *L103 'cumulative surface height change' is (mathematically) equivalent to 'surface height'. I suggest the latter for brevity.*
**AR:** We agree in principle. However, the formulation "observations of ice surface height between two time steps" (as the sentence would read then in l. 103) sounds unintuitive to us. We thus suggest the following wording as a compromise: "observations of surface height change since a given point in time (in our case the time at which the camera is set up)".

**RC:** *Eq. 1 This equation is only valid for bare ice. This is briefly touched on elsewhere, but should be reiterated here. In fact, it might be better to state that the operation relates h(t,z) to $a_{sfc}$.*
**AR:** We suggest to circumvent this issue by replacing the density of ice "$\rho_{ice}$" with the bulk density of snow and/or ice "$\rho_{bulk}$". We think that this should clarify that this equation is generally valid. We will explain in the text that "$\rho_{bulk}$ is the temporally weighted average of the snow and ice densities ($kg\,m^{-3}$) at the camera location.".

**RC:** *L111 'Short snow events ...'. We never see this notion of assigning a high uncertainty SWE estimate again. Is this actually done, and specifically how?*
**AR:** Yes, we actually assign higher uncertainties to snowfalls. We propose to clarify the "how" as follows: "Short snow events during the melt seasons are assigned a density of 150 $kg\,m^{-3}$. The calculated snow water equivalent is assigned an uncertainty of 2-3 cm w.e..".

**RC:** *L133 I'm confused by the lapse rate thing. Why don't you continue to be a Bayesian and just use the probability distribution over the lapse rate inferred from the data without injecting questionable notions of 'significance'? This could then be propagated into downstream analysis.*
**AR:** We did not use significance testing in the strict statistical sense, but rather as a simple procedure to take background information (in the data assimilation sense) into account on days where the lapse rate cannot be estimated from the meteorological data directly. However, we will implement the suggested way to treat the lapse rates of temperature and precipitation, which we understand as follows: we compute the posterior distribution of the lapse rate for each day (using, e.g., a g-prior of Zellner (Zellner, 1986)) and then use an independent draw from this posterior for each particle in the particle filter's predict step. Note that the information required to describe the procedure will most likely result in an increase in the paper's length.

**RC:** *L151 In what sense is an outline a surface? I don't understand this line.*
**AR:** The term "reference surface" is used to highlight that the extent doesn't change over time. We suggest to clarify this in the following way: "[...] mass balances in this study are calculated over a glacier surface area that does not change over time (Elsberg et al., 2001, Huss et al., 2012)."

**RC:** *L158 'Values of glacier-wide mass balance ...' I don't understand what 'partly harmonized' means in this context?*
**AR:** It means that some of the mass balance data used for model calibration (i.e. data provided by GLAMOS), are already consistent with geodetic mass balances, thus matching long-term mass changes

(the procedure of ensuring consistency is often referred to as "homogenization", (Huss et al., 2015)). "Partly homogenized" (or "harmonized") means that this procedure has not yet happened for the most recent mass balance data, since no geodetic mass balances are available yet. We will better explain that in the revised text at l.158 and exchange "harmonized" with "homogenized" to avoid confusion: "Glacier-wide mass balances are obtained by extrapolating the in-situ observations, and making the extrapolated values consistent with long-term mass changes. The latter procedure is often referred to as "homogenization" (e.g. Bauder et al., 2007, Huss et al., 2015). For the recent years this homogenization has not yet been applied, since no geodetic mass balances are available yet."

**RC:** *Eq. 2 Perhaps it's standard notation, but having cprec mean an entirely different thing (with different units) than ccfc is really confusing.*
**AR:** We will replace $c_{\text{prec}}$ with $prcp_{\text{scale}}$ in the entire manuscript to avoid such confusion.

**RC:** *L214 It would be useful to make a note that G is a function of I.*
**AR:** We agree. We will use the following notation in the entire manuscript: $G(I_{\text{pot}}, t, z)$

**RC:** *Eq. 8 Suggest using $\Delta t$ rather than dt, as the latter is usually reserved for infinitesimals.*
**AR:** We will change this as suggested.

**RC:** *Eq. 10 The 'general framework' also has inside epsilon of H(x), although it doesn't appear that way in this work.*
**AR:** We see the point. Our notation, which has $\epsilon$ outside of $\mathcal{H}(x)$, is meant to signalize that we only consider additive errors in our study. This is a common assumption, and it is not obvious to us why considering other error types would be beneficial in our case. However, we agree that some confusion arises due to the different notations in Eqs. 1 and 10. We thus suggest to move $\epsilon_t$ outside of the brackets to match Eq. 10:

$$h(t,z) = \mathcal{H}(b_{\text{sfc}}(t,z)) + \epsilon_{t,z} = \frac{b_{\text{sfc}}(t,z) \cdot \rho_w}{\rho_{\text{bulk}}} + \epsilon_{t,z}$$

**RC:** *Sec. 3.3 I find it confusing that the parameter update process appears at the end, even though Figure 6 indicates that it happens at the same time as the state prediction.*
**AR:** The reason why the two processes are merged in fig. 6 is that we couldn't think of a symbol representing the updating of parameters. In general, the order "resampling - parameter update - prediction" is due to the observations (and so the weights) only depending on the physical state $\xi$, and not on the parameters $\theta$. This is why the parameters can evolve after the resampling, in which it is decided to which models the resampled particles belong. In the example figure, the two orange particles with the same physical state $\xi_t$ can thus obtain different parameters. We suggest to visually separate the parameter evolution from the prediction step by moving the annotation to the resampling side.

[Figure]

**RC:** *L289 It might be clearer to state explicitly that a particle is always associated with only 1 model over its "lifetime".*
**AR:** Thanks for this comment. We suggest to add one more sentence to explain what "a particle" means: "[...], which means that, when following a given particle backwards in time, the entire dynamics of the particle is governed by one single model over its "lifetime". In the other direction, a particle can change model during the resampling step. In this case, both the model index $m_{t,k}$ and the entire past trajectory is changed to the new model.".

**RC:** *Eq. 16 I strongly disagree with the choice of setting $\beta_t = 0$. This is because this is tantamount to the assumption that the model predictions are perfect, which is certainly not the case. In reality, two models are only different in their reliability to the extent that their predictions differ by more than their internal uncertainty and one fits the data better than the other. Setting this model error to zero artificially accentuates the differences in the likelihoods computed for different model and encourages the mode collapse (what you call 'model dominance') exhibited in Figure 11.*
**AR:** Our choice of setting $\beta_t$=0 originates from the fact that for our case, the model error is dominated by the uncertain model inputs, i.e. by the meteorological input and the parameters. In principle, we have thus splitted the value of $\beta$ into the meteorological input errors $\eta$, which we can specify correctly. We suggest to better clarify this point with the following explanation (l. 306): "By introducing both the meteorological input uncertainty $\eta$ and the parameter uncertainties, we shift the majority of the uncertainty contained in $\beta_t$ to these variables. Since the remaining uncertainty for $\beta_t$ is small and hard to quantify, we set $\beta_t$=0 for simplicity."

**RC:** *Table 2 Caption By covariance, do you mean standard deviation?*
**AR:** Yes, sorry, that's a typo.

**RC:** *L320 the standard symbol for variance would be $\sigma^2$.*
**AR:** For clarity (one superscript less), we will change "variance" to "standard deviation" and leave the symbol $\sigma_\epsilon$ untouched.

**RC:** *Eq. 18 An implicit assumption made throughout is that a single model's probability is marginally uniform, or alternatively that $P(m_t) = Dirichlet(1)$, to wit that one model being dominant is just as probable as all four models contributing equally. This is a weird assumption for a time dependent problem, because it means that physical reality is subject to sudden switches between governing principles. Again, this leads to the mode collapse seen in Fig. 11. Predictions might be made substantially more robust by putting a prior on P(mt) such that the more probable case is an averaging of the four models, and deviation from that has to be the result of significant evidence.*
**AR:** We do not make the assumption that the probability of each single model is marginally uniform, neither explicitly nor implicitly. At the initial time $t_0$, all four models have the probability 1/4. As the model index never changes during predict steps, the same is true at all later times $t$ if we do not condition on the observations. If we condition on the observations, instead, the model probabilities change at each time a new observation becomes available (this happens during the update step). In other words, we have

$$P(m_t = j \mid y_{1:t-1}) = P(m_{t-1} = j \mid y_{1:t-1}),$$

This means that a model that has high probability at time $t - 1$ is favored by the prior also at time $t$, which means that a sudden switch of the preferred model is unlikely. Such a switch only happens if the evidence for it is strong, the evidence being given by the likelihood ratio and the likelihood of model $j$ being

$$p(y_t \mid m_t = j, y_{1:t-1}) \propto \frac{p(m_t = j \mid y_{1:t})}{p(m_t = j \mid y_{1:t-1})},$$

or, by Equations (17) and (18),

$$p(y_t \mid m_t = j, y_{1:t-1}) \propto \frac{\sum_k p(y_t \mid x_{t,k}) w_{t-1,k} \delta(m_{t,k} - j)}{\sum_k w_{t-1,k} \delta(m_{t-1,k} - j)}.$$

The preferred model can thus only switch if the particles belonging to one model have a much better fit for the new observations than all the other models. This is what happens with our data. It indicates that presumably the forecast values $x_{t,k}$ are overconfident and that all four models have non-negligible model errors. However, in order to specify model errors, we would need physical knowledge of their

order of magnitude and their dependence on meteorological inputs. As a way to mirror this explanation in the manuscript, we suggest to show a figure containing the prediction $b_{sfc}(z,t)_k$ and the likelihood $p(y_t \mid b_{sfc}(z,t))$ for a time $t$ where the model probability changes quickly.

**RC:** *L338 It's not that there's no stochasticity, it's that mt for a given particle doesn't evolve at all!*
**AR:** Thanks for raising this point. We think that the statement depends on how the system is viewed and thus how a given particle is defined. When taking the Eulerian view, i.e. when the particle index $k$ is fixed, the model index $m_{t,k}$ is free to change during resampling. When taking the Lagrangian view, i.e. when not fixing $k$ but following a specific particle backwards in time, its model index does not change (see also our response to the comment on line 289). In this sense, we believe that our statement "There is no stochasticity in the evolution of $m_t$" is correct. To clarify this, we suggest to rephrase l. 338 in the following way: "Because there is no stochasticity in the evolution of $m_t$ though, when the particle index $k$ is fixed,..."

**RC:** *Sec. 3.4 This section is essentially incomprehensible, with the section on proper scoring reading like it was pasted from a statistical methods paper. This being the Cryosphere, it's important to try to help your reader with some intuition as to what the CRPS actually means, and why its potential impropriety matters. A figure describing the metric might be useful, or perhaps a simple example de-scribing circumstances where the value is high or low. While the rest of the paper is still accessible not understanding CRPS, the analysis breaks down to 'big number bad, low number good,' which is unfortunate given that there is probably much more insight to be gained from the following sections.*
**AR:** We suggest to simplify the paragraph and take some statistics jargon out of it. We will also add the suggested example (but prefer not to add the figure for not increasing the manuscript's length further): "[...]It takes into account both the deviation of the median forecast from the actual observation (forecast reliability) and the spread of the forecast distribution (forecast resolution). This means that a forecast close to the observation median can receive a poor Continuous Ranked Probability Score (CRPS) if the forecast distribution spread is high, and the other way around. The CRPS is defined as (Hersbach, 2000): [...]Lower values of CRPS correspond to better forecasts. The minimum value is zero, corresponding to a deterministic, perfect forecast." We will introduce further edits to the explanation of non-proper and proper CRPS in the revised manuscript.

**RC:** *Sec. 4.2.1 This section is quite unclear, specifically what the differences are that these include relative to the 'full' forecast.*
**AR:** We want to avoid the wording "full" in this paragraph and suggest to change the sentences into: "We have run experiments where the particle filter is limited to using mean parameters and/or single models instead of parameter distributions and the model ensemble. In more than half of the experiments, the resulting CRPS values are higher than the highest CRPS obtained with the ensemble setting and time-variant parameters."

**RC:** *L435 Perhaps I missed it, but I can't find anything describing what the number in brackets means.*
**AR:** It is explained in l. 371, but indeed far from the first number occurring in this format. We will add another hint in the Results section (see also comment from Anonymous Referee #2 on this line): "(proper CRPS outside, non-proper CRPS inside the square brackets)".

**RC:** *Sec. 4.2.2 This section on cross-validation is very clear and good. Maybe it would be useful to comment on the temporal pattern evident in Figure 9, with CRPS in-creasing through time, but at different rates between different cross-validation folds.*
**AR:** This is indeed a very interesting feature. We suggest to add the following sentences: "The temporal pattern evident in Figure 9 includes an increasing CRPS through time, but at different rates between the individual cross-validation folds. It originates from (1) how representative camera stations are for the elevation band they are located in, (2) how the stations are combined in the cross-validation folds, and (3) the cumulative error characteristics, since we observe cumulative mass balance over time. To mention an example, station RHO 3 can generally be modeled with low errors compared to other stations. This is because the station is located in reasonably flat terrain with only little crevasses. The other stations are instead either in the vicinity of crevasses (RHO 4) or influenced by shadows from the surrounding terrain, dark glacier surface or steep ice (RHO 1 and RHO 2). RHO 1 and RHO 2 show that also neighboring stations can exhibit different melt, leading to a different reproducibility in the cross-validation."

**RC:** *L481 I don't understand where the '45 distinct model runs' come from. Also, what is a 'random coupling'?*

**AR:** We suggest to clarify this in the text in the following way: "The 45 model runs come from the 45 parameter sets we could gain from the calibration described in section 3.2. The random coupling is a random connection between the 45 model runs and the 10000 particle trajectories that initiate when the particle filter run starts."

**RC:** *L495–496 I don't understand this sentence, nor why conditioning initial conditions on observations leads to poorer results.*

**AR:** This is something that we discussed extensively as well. The only reasons we can think of for why the conditioning can also lead to worse overall results are the following: 1) the mass balance stakes are several meters to hundreds of meters away from the camera installations, and are thus not "true" observations at the camera locations. This might result in the initial conditions being conditioned on biased observations with respect to the camera locations. 2) Either the CRAMPON and/or the GLAMOS uncertainties might be too small or too large. This can cause either of the analyses to be over-confident. 3) The GLAMOS glacier-wide annual mass balances are interpolated from the stake readings using a model, and CRAMPON uses simplified geometries to calculate mass balances. If combined unluckily, this might lead to a stronger deviation from the GLAMOS annual mass balance when conditioned on point observations. Concretely, this means that because it was only possible to mount the cameras in our study on the lower 30% of the glacier surface area, they have a spatial bias and thus might not be representative for the vertical mass balance gradient. The three points are mentioned in the manuscript already, but we would like to elaborate more on point (2) and (3): "Overall differences to the GLAMOS analyses can be explained by [. . .], (2) the use of only 1-4 point observations located in the ablation zone and covering max. 30% of the glacier surface , compared to the complete network of 5-14 stakes over the entire elevation range used in the GLAMOS analyses, (3) lack of representativeness of the camera observations for the accumulation zone of the glaciers, i.e. biased vertical mass balance gradients, [. . .]"

**RC:** *Figure 11 To emphasize earlier comments again, this pattern of mode collapse is strongly indicative of an over-confident likelihood operating in an M-open framework. It's well known that Bayesian inference only 'works' when the models are correctly specified. For Bayesian model averaging (which is what the particle filter is doing in a time dependent way), this still holds: because the true physics are not contained in the set of equations that the filter has available to pick from, yet this additional uncertainty is not explicitly specified, the filter hops between the model that fits the observations in the moment. While I don't expect any additional analysis, I think it would be appropriate to make this assumption explicit in the text, and to perhaps reference it when describing the fast switching between dominant models.*

**AR:** This is really a valuable inspiration for future work. We agree with the comment but would add that, in our view, it is not the likelihood being overconfident, but rather the prediction and thus the prior following from that. This is because we have chosen the observational error $\sigma_\epsilon$ conservatively. We could experiment with likely values for $\beta_t$, which would also depend on the meteorological input uncertainties, until the Particle Filter slowly converges towards one model. However, we believe that this is critical, since values for $\beta_t$ are volatile and tuning them would be subject to manual intervention. When implementing the procedure suggested for the lapse rates as our reply to the comment on l. 133, we will take another source of uncertainty into account, which counteracts the "hopping" between models. In order to include an explicit error term $\beta_t$, we would have to specify a distribution also depending on the meteorological inputs. This would be a major undertaking. Regarding the request to clarify this in the text, we propose the following addition (together with the edits according to the comment on Eq. 16): L.307 "By introducing both the meteorological input uncertainty $\eta$ and the parameter uncertainties, we shift the majority of the uncertainty contained in $\beta_t$ to these variables. Since the remaining uncertainties for $\beta_t$ are small and hard to quantify, we set $\beta_t=0$ for simplicity. With this assumption we neglect some additional uncertainty contained in $\beta_t$, which might lead to jumps in the temporal evolution of the model probability."
L. 515 "This model dominance, and especially the fast switches between dominant models, is describing a mode collapse. This might indicate an overconfident likelihood and/or a prior operating in an M-open framework (Bernardo and Smith, 2009) where the "true" model is not a choice amongst the available models. In our case, we believe that the ensemble prior is overconfident, since we have chosen the observational error conservatively. The filter thus switches back and forth quickly between individual models that describe the observations best. We accept the fast switching model dominance as a sign that the overall ensemble performance is improved."

**RC:** *Figure ?? Two things that are missing from the paper are time series' of state and parameter distributions. It would be very interesting to see the evolution of un-certainty in the predictions away from observations, and also to see how quickly parameters change or revert to the mean.*

**AR:** We agree that this is very interesting. We will add a figure and a small paragraph discussing this aspect in the new manuscript.

**New references**

- Bernardo, J. M., & Smith, A. F. (2009). Bayesian theory (Vol. 405). John Wiley & Sons.

- Zellner, A. (1986). On assessing prior distributions and Bayesian regression analysis with g-prior distributions. Bayesian inference and decision techniques.

---

## Author Comment (AC2) · 19 Jun 2021

Dear Anonymous Referee #2,

Dear Editor,

We would like to thank Anonymous Referee #2 for the valuable review of our manuscript. We appreciate the constructive comments and the positive feedback regarding the overall context of the study. We attach a supplementary file with point-by-point answers to the individual remarks and questions.

Best regards,

Johannes Landmann and co-authors

Please also note the supplement to this comment:
https://tc.copernicus.org/preprints/tc-2020-281/tc-2020-281-AC2-supplement.pdf

───────────────────────────────

**Supplement:**

**Author response to the review of Anonymous Reviewer #2**

Johannes Landmann and co-authors

June 2021

Dear Anonymous Referee #2,
Dear Editor,
We would like to thank Anonymous Referee #2 for the valuable review of our manuscript. We appreciate the constructive comments and the positive feedback regarding the overall context of the study. We address the comments point-by-point answers to the individual remarks and questions.
Best regards,
Johannes Landmann and co-authors

**– Major comments and questions –**

***RC:*** *What made you choose a particle filter as DA method as opposed to more traditional methods (e.g. variational methods and ensemble Kalman filter). It would complement the work if you discussed why a PF suits this problem.*
**AR:** The Particle Filter (PF) is a generic data assimilation method that can handle all kinds of model distributions, also non-linear ones. We have chosen the PF since we know that the distributions we deal with are not always Gaussian. To not extend the already long introduction any further, we suggest to add these explanations at l. 250, where we introduce the Particle Filter: "Especially when temperatures are around the melting point, the system becomes non-linear since melt occurs above but not below this point. As a consequence, the distributions we deal with are not necessarily Gaussian. The facts that (a) the temperature chosen to parametrize the melting point is not the same for all four models, (b) the individual model prior distributions are combined to obtain the ensemble prediction, and (c) there can also be accumulation contributing to the overall mass balance, add further complexity. We do not use other data assimilation approaches, such as variational methods or Ensemble Kalman filtering, because variational methods encounter difficulties when dealing with non-Gaussian priors (van Leeuwen et al., 2019), whilst the Ensemble Kalman Filter in its original form is not designed for multi-model applications as we use in our case. Overall, particle filtering is a very flexible, generalizable, and readily implementable data assimilation method."

***RC:*** *Line 105. If I understood well, the observations are of a cumulative quantity. In this case, do observation errors need to consider time auto-correlations?*
**AR:** This is a justified question. No, the observations are not cumulative, in the sense that the mass balance at a given time is not inferred by summing the individual, sub-daily readings up to that time. Rather, the mass balance of a given point in time is given by one single reading at that time. In this sense, the individual measurements are independent, and only affected by the precision by which we can read a stake at a given moment. To clarify this, we suggest adding the following text at L.107: "We expect the observation errors to be uncorrelated in time, since every reading is independent from the previous one."

***RC:*** *Line 255.I was a bit confused on where the uncertainties of the input variables are represented. Are they represented in the model error beta, in the observation error epsilon (as mentioned in line 258), or both?*
**AR:** The observational error from the camera readings is represented in the observation error $\epsilon$, as stated in l. 257. The model input errors are considered to be contained in the model error $\beta_t$, as stated in l. 258. We suggest to better clarify this by replacing "[. . . ] $(\beta_t)$ can also represent uncertainties in model input variables" with "[. . . ] $(\beta_t)$ should include the uncertainties about model input variables" in l. 258.

***RC:*** *The use of the PF in a multi-model ensemble context is quite interesting, especially since each model has different parameters one is trying to estimate. Is there previous work in this regard? Could you*

*provide some references?*

**AR:** Currently, we are not aware of other studies that have applied particle filtering in a multi-model ensemble related to glacier mass balance. However, we have added some references to applications in other contexts:

- Kreucher, C., Hero, A., & Kastella, K. (2004, March). Multiple model particle filtering for multitarget tracking. In Proceedings of the Twelfth Annual Workshop on Adaptive Sensor Array Processing.

- Ristic, B., Arulampalam, S., & Gordon, N. (2004). Beyond the Kalman filter: Particle filters for tracking applications (Vol. 685). Boston: Artech house.

- A. Saucan, T. Chonavel, C. Sintes and J. Le Caillec, "Interacting multiple model particle filters for side scan bathymetry," 2013 MTS/IEEE OCEANS - Bergen, 2013, pp. 1-5, doi: 10.1109/OCEANS-Bergen.2013.6608125.

- Wang, R., Work, D. B., & Sowers, R. (2016). Multiple model particle filter for traffic estimation and incident detection. IEEE Transactions on Intelligent Transportation Systems, 17(12), 3461-3470.

The revised text will read: "We are not aware of mass balance studies that have applied a multi-model ensemble based on a particle filter with the resampling methods we propose, although multi-model particle filters have been used for other applications (e.g. Kreucher et al., 2004, Ristic et al., 2004, Saucan et al., 2013, Wang et al., 2016)."

*RC: When discussing the particle filter, you introduce the concept of 'minimum contribution' for some particles. This is taken into account when weighting, as explained in appendix 2. There is a comment saying that the original weights are preserved 'in average'. Could you elaborate more on this statement?*

**AR:** Only preserving the weights on average is common for resampling procedures: when a particle performs poorly, it obtains a weight of zero and disappears. If a particle $x_{t,k}$ has a weight $w_{t,k}$, then it is chosen $N \cdot w_{t,k}$ times in the resampling. This particle then has the weight $1/N$ times the "number how often it has been resampled", so on average $w_{t,k}$. The same is true when resampling within a model: a particle $x_{t,k}$ with model index $j$ is resampled on average $N_{t,j} \cdot w_{t,k}/\pi_{t,j}$ times. After resampling it has the weight $\tilde{w}_{t,k}$ times "number how often it was resampled", so on average $w_{t,k}$. This statement is made in Equations B2 and B3, which are found in Appendix B.

*RC: Equation 21. How are $\mu_0$ and $\Sigma_0$ chosen?*

**AR:** We choose $\mu_0$ and $\Sigma_0$ from the parameter statistics obtained from the calibration procedure in section 3.2. We suggest to add a phrase stating where $\mu_0$ and $\Sigma_0$ originate from. Suggested revised text: "$\vec{\mu}_0$ and $\vec{\Sigma}_0$ are the prior mean and the prior covariance of $\vec{\theta}$ at the starting time $t_0$, which we determine from the calibration procedure described in section 3.2,[...] "

– **Minor comments and questions** –

*RC: The title mentions 'mass balance observations', whereas the observations are of surface elevation.*
**AR:** Our intention was to simplify the wording for the reader. As explained in our response to the comment on the manuscript title by Reviewer #1, we suggest to change the title as follows: "Assimilating near real-time mass balance stake readings into a model ensemble using a particle filter"

*RC: Line 96. It is mentioned that the camera images are read 'manually' to obtain the daily cumulative surface height change. Is it literally reading the marks from the ablation pole? How could this be automated to be applied to more places?*
**AR:** Yes, we have read the marks manually. We are working on a procedure to automate this though, so that operational runs that we plan for the future won't require manual interventions. In this respect, see our EGU2021 abstract (https://doi.org/10.5194/egusphere-egu21-7663) and our GitHub repository (https://github.com/leosold/TOAST).

*RC: Line 127. Can you say more about the 0.2 degree resolution? How does this compare with other products? Is it high or low resolution?*
**AR:** At Swiss latitudes, 0.2 degrees corresponds to a resolution of about 2km. Compared to Global Climate Models or Regional Climate Models, which are sometimes used to force glaciological models

directly, this is a very high resolution. We will specify this in the revised text: ".... which for Switzerland corresponds to a horizontal resolution of about 2 km."

**RC:** *Figure 7. I think making the vertical axis larger for panels a and b could make the figure easier to read.*
**AR:** As a response to the request, we suggest to double the extent of these two axes:

[Figure]

**RC:** *Figure 8. The individual circles are difficult to see. Please make the circumferences thicker, and maybe increase the size of the figure.*
**AR:** We will increase the line thickness of the circles to improve the visibility and also increase the figure size:

[Figure]

**RC:** *In pages 22-24 (approximately) there are several places where a quantity is written followed by () or []. It was not clear to me what the quantities in the parenthesis are, and why there are two styles.*
**AR:** It was explained in l. 371 that we use the square brackets to add the non-proper Continuous Ranked Probability Score (CRPS) into the text, but apparently it needs to be repeated in the Results section. Moreover, the round brackets we use in our notation are to be understood as annotations in the way

they are commonly used. We will add a reminder at the first occurrence of the square brackets: "(proper CRPS outside, non-proper CRPS inside the square brackets)".

**- Typos and corrections -**

**RC:** *Line 109. ... because it can happen that the camera construction sinks...→ ... because the camera construction can sink. (easier to read).*
**AR:** We will change as suggested.

**RC:** *Line 118. melt during night → nighttime melting*
**AR:** We will change this as suggested.

**RC:** *Line 190 on. When mentioning the models in an itemised list, start the sentences with capital letter.*
**AR:** We will change this as suggested.

**RC:** *Figure 5. Some of the words in the labels are split into two lines*
**AR:** This was a tradeoff between font size and visual appearance. We will reformat the figure so that the individual words are better readable:

[Figure]

**RC:** *In the title of table 2 it should say 'standard deviations' instead of 'covariances'.*
**AR:** We thank the reviewer for noticing this. It will be corrected.

---

## Referee Report (RR1)

**Referee report on 'Assimilating near real-time mass balance stake readings into a model ensemble using a particle filter'**

Doug Brinkerhoff

October 2021

**Summary**

In this manuscript, the authors develop a Bayesian method for incorporating photography-based observations of glacier melt into a multi-model ensemble of surface mass balance models via a particle filter. They show that different models fit the observations at different times and that their method is able to select between the different models through time. They find that the method performs very well in predicting cumulative mass balance, but more importantly, that it comes with robust estimates of uncertainty.

**Major Comments**

I reviewed this paper once before and suggested a variety of changes and clarifications. The authors have done an exceptional job in incorporating these suggestions, to the extent that I have not been able to identify anything I particularly disagree with in the new version. I find this manuscript to be well-written, scientifically sound, and a very interesting contribution to mass balance modelling specifically, and Bayesian methods in glaciology more generally.

**Technical Corrections**

**L308–309** This is semantics, but the particle doesn't change model during the resampling step: it 'dies off' and is replaced by another particle.

---

## Author Response (AR2)

**Author response to the review of D. Brinkerhoff**

Johannes Landmann and co-authors

June 2021

Dear Douglas Brinkerhoff,
Dear Editor,
We thank Douglas Brinkerhoff very much for the valuable comments regarding our manuscript. We really appreciate the thoughtful input and the detailed recommendations to improve the document. We address the raised requests in the form of point-by-point responses and explain how we updated the final manuscript.
We have also taken the opportunity to further streamline out wording, by simplifying a number of sentences and paying attention to the overall text coherence. We are convinced that this has resulted in a much more accessible manuscript as compared to the first submission.
Best regards,
Johannes Landmann and co-authors

**RC:** *Title The observations are not of mass balance, but of surface elevation (specifically in the negative direction). I suggest changing the title to be more precise.*
**AR:** Thanks for this comment. We agree that the observations we make are in "surface elevation space" and that these are then transformed into a mass change using the density of the medium (snow/ice). However, the reviewer will agree that (i) the surface elevation change of a glacier at a given point is the result of both surface mass balance and a component due to ice emergence (see Cogley et al., 2011, p. 38), and (ii) the changes that we measure at our stakes are only due to the former. Stated differently: what we see is the surface elevation change due to glacier mass balance, but not the total ice thickness change. To avoid possible confusion, we have changed the title to: "Assimilating near real-time mass balance stake readings into a model ensemble using a particle filter".

**RC:** *L10 The reader does not yet know what 'model probability' is in the abstract, nor is the abstract notion of 'custom resampling' useful here.*
**AR:** We understand the point and we have written the following, simplified text: "By means of a particle filter, these observations are assimilated into an ensemble of three temperature index (TI) and one simplified energy-balance mass balance models. State augmentation with model parameters is used to assign temporally-varying weights to individual models. We analyse model performance over the observation period, and find that the probability for a given model to be preferred by our procedure is [...]".

**RC:** *L39 Of the three points (first, second, third) made after this line, only one logically follows this statement.*
**AR:** We agree that the logics might not have been apparent because the enumerations were stretched very far out into the subsequent text. We have restructured the text as follows: "In many cases, mass balance analyses are available twice a year, and are based on seasonal in situ observations (Cogley et al., 2011). This relatively low frequency is related to the fact that in situ observations are expensive in terms of both time and manpower. Only recently have low-cost and high-frequency monitoring approaches emerged (Hulth, 2010, Fausto et al., 2012, Keeler and Brugger, 2012, Biron and Rabatel, 2019, Carturan et al., 2019, Gugerli et al., 2019, Netto and Arigony-Neto, 2019). However, even with these observations, it is not straightforward to provide analyses at higher frequencies. This is because near real-time estimates are often based on ensemble modelling, in order to enable a correct quantification of uncertainties. Ensemble modelling is used in glaciology in the context of model intercomparison projects (Hock et al., 2019), future projections for ice sheets and mountain glaciers (e.g. Ritz et al., 2015, Shannon et al., 2019, Golledge, 2020, Marzeion et al., 2020, Seroussi et al., 2020), and also to determine the initial conditions for modelling (Eis et al., 2019). However, ensembles are currently not prominent in the calculation of seasonal or daily glacier mass balances. Another reason why calculating higher-frequent glacier mass balance analyses is

not straightforward is the lack of knowledge about the short-term variability in the parameters of the necessary models. temperature index (TI) models, for example, are parametrizations of the full energy balance equation and offset some of the changes occurring in the driving processes through parameter fluctuations (Ohmura, 2001, Lang and Braun, 1990, Hock, 2003, Hock et al., 2005). In a comparison of four TI models and a full energy balance model, Gabbi et al. (2014) showed that all models perform very similarly on a multi-year scale.".

**RC:** *L49 List of references should have an e.g. in front of it. There are many other examples of ensemble modelling for ice sheet projection.*
**AR:** Thanks! We have inserted "e.g.".

**RC:** *L55 'discussed how'→'not clear whether'(?)*
**AR:** We have exchanged the wording as proposed.

**RC:** *L63 surface point mass balance→surface point ablation. You don't measure massbalance, you measure volume change in one direction.*
**AR:** Besides the answer given in reply to the comment related to the title (see our first answer), note that we do not only measure ablation but also accumulation. Together, this is the mass balance of the surface.

**RC:** *L80 as above.*
**AR:** Idem.

**RC:** *L103 'cumulative surface height change' is (mathematically) equivalent to 'surface height'. I suggest the latter for brevity.*
**AR:** We agree in principle. However, the formulation "observations of ice surface height between two time steps" (as the sentence would read then in l. 103) sounds unintuitive to us. We have inserted the following wording as a compromise: "observations of surface height change since an initial point in time (in our case the time at which a camera is set up)".

**RC:** *Eq. 1 This equation is only valid for bare ice. This is briefly touched on elsewhere, but should be reiterated here. In fact, it might be better to state that the operation relates h(t,z) to $a_{sfc}$.*
**AR:** We have circumvented this issue by replacing the density of ice "$\rho_{ice}$" with the bulk density of snow and/or ice "$\rho_{bulk}$". We think that this clarifies that this equation is generally valid. We now explain in the text that "$\rho_{\text{bulk}}$ is the temporally weighted bulk density of snow and ice at the camera location $(\text{kg m}^{-3})$".

**RC:** *L111 'Short snow events ...'. We never see this notion of assigning a high uncertainty SWE estimate again. Is this actually done, and specifically how?*
**AR:** Yes, we actually assign higher uncertainties to snowfalls. We have clarified the "how" as follows: "Short snow events during the melt season are assigned a density of 150 $\text{kg m}^{-3}$. The calculated snow water equivalent is assigned an uncertainty of 2-3 cm w.e.."

**RC:** *L133 I'm confused by the lapse rate thing. Why don't you continue to be a Bayesian and just use the probability distribution over the lapse rate inferred from the data without injecting questionable notions of 'significance'? This could then be propagated into downstream analysis.*
**AR:** We did not use significance testing in the strict statistical sense, but rather as a simple procedure to take background information (in the data assimilation sense) into account on days where the lapse rate cannot be estimated from the meteorological data directly. However, we have implemented the suggested way to treat the lapse rates of temperature and precipitation in the following way and explained it in the text: The temperature lapse rate is now derived from the 25 closest cells to a glacier outline centroid using a Bayesian estimation based on a linear regression model:

$$T_{t,i} = e_t + q_t h_i + \nu_{t,i} \tag{1}$$

where $T_{t,i}$ is the temperature of the i-th grid cell out of the 25 considered cells at time $t$, $e_t$ is the regression line intercept, $q_t$ is the regression slope (i.e. $\frac{\partial T}{\partial z}$), $h_i$ is the height of the i-th grid cell, and $\nu_{t,i} \sim \mathcal{N}(0, \sigma_{\nu,t}^2)$ are the residuals independent in space and time. Using a g-prior of Zellner (Zellner, 1986), being non-informative in the intercept $e_t$ and model noise variance $\sigma_{\nu,t}^2$ of the regression, we draw samples of the lapse rate $q_t$ from the following posterior distribution:

$$p(q_t \mid T_t) \propto \left(1 + \frac{\left(q_t - \frac{g}{1+g}\hat{q}_t - \frac{1}{1+g}q_0\right)^2}{24c^2}\right)^{-25/2} \tag{2}$$

with

$$c^2 = \frac{g}{24(1+g)\sum(h_i - \bar{h})^2}\left(s_t^2 + \frac{1}{1+g}\sum_i(h_i - \bar{h})^2(\hat{q}_t - q_0)^2\right). \tag{3}$$

Above, $p(\cdot)$ means "probability of", $g$ determines a weighting factor composing the posterior mean (we set $g = 1$), $\hat{q}_t$ is the least squares estimator of the slope, $q_0$ is the prior mean, which we choose to be an annually varying climatological mean gradient at the respective grid location, $\bar{h}$ is the average height of the 25 grid cells, and $s_t^2$ is the residual sum of squares. This is up to a constant the density of a $t$-distributed random variables with 24 degrees of freedom, shifted by $\frac{(g\hat{q}_t + q_0)}{(1+g)}$ and multiplied by $c$. The samples drawn from this distribution are then propagated into the particle filter. For precipitation, we also derive Bayesian lapse rates from the surrounding 25 grid cells in the same fashion as we do for the temperature lapse rate. However, to circumvent high errors in the slope calculation due to the boundedness of precipitation towards zero, we (1) calculate the slope on the square root of the precipitation, and (2) assign a probability that the reference has actually received precipitation when the reference cell value is zero but other cells have non-zero precipitation.

**RC:** *L151 In what sense is an outline a surface? I don't understand this line.*
**AR:** The term "reference surface" is used to highlight that the extent doesn't change over time. We have clarified this in the following way: "[...] and mass balances are calculated over a fixed glacier surface area (Elsberg et al., 2001, Huss et al., 2012)"

**RC:** *L158 'Values of glacier-wide mass balance ...' I don't understand what 'partly harmonized' means in this context?*
**AR:** It means that some of the mass balance data used for model calibration (i.e. data provided by GLAMOS), are already consistent with geodetic mass balances, thus matching long-term mass changes (the procedure of ensuring consistency is often referred to as "homogenization", (Huss et al., 2015)). "Partly homogenized" (or "harmonized") means that this procedure has not yet happened for the most recent mass balance data, since no geodetic mass balances are available yet. We have explained that in the revised text at l.158 and exchange "harmonized" with "homogenized" to avoid confusion: "Glacier-wide mass balances are obtained by extrapolating the in-situ observations, and making the extrapolated values consistent with long-term mass changes. The latter procedure is sometimes referred to as "homogenization" (e.g. Bauder et al., 2007, Huss et al., 2015). For recent years, this homogenization has not yet been performed, since no geodetic mass balances are available yet."

**RC:** *Eq. 2 Perhaps it's standard notation, but having cprec mean an entirely different thing (with different units) than ccfc is really confusing.*
**AR:** We have replaced $c_{\text{prec}}$ with $prcp_{\text{scale}}$ in the entire manuscript to avoid such confusion.

**RC:** *L214 It would be useful to make a note that G is a function of I.*
**AR:** We agree. We use the following notation in the entire manuscript now: $G(I_{\text{pot}}, t, z)$

**RC:** *Eq. 8 Suggest using $\Delta t$ rather than dt, as the latter is usually reserved for infinitesimals.*
**AR:** We have changed this as suggested.

**RC:** *Eq. 10 The 'general framework' also has inside epsilon of H(x), although it doesn't appear that way in this work.*
**AR:** We see the point. Our notation, which has $\epsilon$ outside of $\mathcal{H}(x)$, is meant to signalize that we only consider additive errors in our study. This is a common assumption, and it is not obvious to us why considering other error types would be beneficial in our case. However, we agree that some confusion arises due to the different notations in Eqs. 1 and 10. We have thus moved $\epsilon_t$ outside of the brackets to match Eq. 10:

$$h(t, z) = \mathcal{H}(b_{\text{sfc}}(t, z)) + \epsilon_{t,z} = \frac{b_{\text{sfc}}(t, z) \cdot \rho_w}{\rho_{\text{bulk}}} + \epsilon_{t,z}$$

*RC: Sec. 3.3 I find it confusing that the parameter update process appears at the end, even though Figure 6 indicates that it happens at the same time as the state prediction.*

**AR:** The reason why the two processes are merged in fig. 6 is that we couldn't think of a symbol representing the updating of parameters. In general, the order "resampling - parameter update - prediction" is due to the observations (and so the weights) only depending on the physical state $\xi$, and not on the parameters $\theta$. This is why the parameters can evolve after the resampling, in which it is decided to which models the resampled particles belong. In the example figure, the two orange particles with the same physical state $\xi_t$ can thus obtain different parameters. We have visually separated the parameter evolution from the prediction step by moving the annotation to the resampling side:

[Figure]

Figure 1: Illustration of the individual particle filter steps. The example refers to a case in which four models (blue, orange, red, and green) start with two particles each. The blue curve represents the observation distribution. At time step $t_0 + 1$, the green model performs poorly and receives entirely low weights during the update step (weights are shown by the size of the circles). In the resampling step, we modify the weights of the other particles again. This is for not omitting the green model entirely, due to temporarily poor performance. As the green model stays in the ensemble, it can recover later, i.e. when making a good prediction (here: $t_0 + 2$).

*RC: L289 It might be clearer to state explicitly that a particle is always associated with only 1 model over its "lifetime".*

**AR:** Thanks for this comment. We have added text to explain what "a particle" means: "This means that, when following a given particle backwards in time, its entire dynamics is governed by one single model only. In the forward direction, a particle can change model during the resampling step. In this case, both the model index $m_{t,k}$ and the entire past trajectory are changed to the new model.".

*RC: Eq. 16 I strongly disagree with the choice of setting $\beta_t = 0$. This is because this is tantamount to the assumption that the model predictions are perfect, which is certainly not the case. In reality, two models are only different in their reliability to the extent that their predictions differ by more than their internal uncertainty and one fits the data better than the other. Setting this model error to zero artificially accentuates the differences in the likelihoods computed for different model and encourages the mode collapse (what you call 'model dominance') exhibited in Figure 11.*

**AR:** Our choice of setting $\beta_t$=0 originates from the fact that for our case, the model error is dominated by the uncertain model inputs, i.e. by the meteorological input and the parameters. In principle, we have

thus splitted the value of $\beta$ into the meteorological input errors $\eta$, which we can specify correctly. We have clarified this point with the following explanation (l. 306): "By introducing both the meteorological uncertainty $\eta$ and the parameter uncertainties, we shift the majority of the uncertainty contained in $\beta_t$ to these variables. Since the remaining uncertainty for $\beta_t$ is small and hard to quantify, we set $\beta_t = 0$ for simplicity."

**RC:** *Table 2 Caption By covariance, do you mean standard deviation?*
**AR:** Yes, sorry, that's a typo. We have corrected it.

**RC:** *L320 the standard symbol for variance would be $\sigma^2$.*
**AR:** For clarity (one superscript less), we have changed "variance" to "standard deviation" and left the symbol $\sigma_\epsilon$ untouched.

**RC:** *Eq. 18 An implicit assumption made throughout is that a single model's probability is marginally uniform, or alternatively that $P(m_t) = Dirichlet(1)$, to wit that one model being dominant is just as probable as all four models contributing equally. This is a weird assumption for a time dependent problem, because it means that physical reality is subject to sudden switches between governing principles. Again, this leads to the mode collapse seen in Fig. 11. Predictions might be made substantially more robust by putting a prior on P(mt) such that the more probable case is an averaging of the four models, and deviation from that has to be the result of significant evidence.*
**AR:** We do not make the assumption that the probability of each single model is marginally uniform, neither explicitly nor implicitly. At the initial time $t_0$, all four models have the probability 1/4. As the model index never changes during predict steps, the same is true at all later times $t$ if we do not condition on the observations. If we condition on the observations, instead, the model probabilities change at each time a new observation becomes available (this happens during the update step). In other words, we have

$$P(m_t = j \mid y_{1:t-1}) = P(m_{t-1} = j \mid y_{1:t-1}),$$

This means that a model that has high probability at time $t - 1$ is favored by the prior also at time $t$, which means that a sudden switch of the preferred model is unlikely. Such a switch only happens if the evidence for it is strong, the evidence being given by the likelihood ratio and the likelihood of model $j$ being

$$p(y_t \mid m_t = j, y_{1:t-1}) \propto \frac{p(m_t = j \mid y_{1:t})}{p(m_t = j \mid y_{1:t-1})},$$

or, by Equations (17) and (18),

$$p(y_t \mid m_t = j, y_{1:t-1}) \propto \frac{\sum_k p(y_t \mid x_{t,k}) w_{t-1,k} \delta(m_{t,k} - j)}{\sum_k w_{t-1,k} \delta(m_{t-1,k} - j)}.$$

The preferred model can thus only switch if the particles belonging to one model have a much better fit for the new observations than all the other models. This is what happens with our data. It indicates that presumably the forecast values $x_{t,k}$ are overconfident and that all four models have non-negligible model errors. However, in order to specify model errors, we would need physical knowledge of their order of magnitude and their dependence on meteorological inputs. As a way to mirror this explanation in the manuscript, we have added a figure containing the prediction $b_{sfc}(z, t)_k$ and the likelihood $p(y_t \mid b_{sfc}(z, t))$ for a time $t$ where the model probability changes quickly (see Fig. 2). We have added the following text: "In terms of the temporal evolution, the model dominance for Rhonegletscher and Glacier de la Plaine Morte is determined already within the first few days, and changes only little after that. Changes in model dominance can be observed for Rhonegletscher and Findelgletscher, instead. In the case of Rhonegletscher, for example, the model dominance switches from the PelicciottiModel to the OerlemansModel and later to the HockModel. For Findelgletscher instead, there is a transition from the OerlemansModel to the PelicciottiModel. This transition is particularly noticeable between August 7th and 8th, 2019 (Fig. 2). The causes for it are not entirely clear, and we speculate that it might be related to the precipitation event starting on August 6th."

**RC:** *L338 It's not that there's no stochasticity, it's that mt for a given particle doesn't evolve at all!*
**AR:** Thanks for raising this point. We think that the statement depends on how the system is viewed and thus how a given particle is defined. When taking the Eulerian view, i.e. when the particle index $k$ is fixed, the model index $m_{t,k}$ is free to change during resampling. When taking the Lagrangian view, i.e.

[Figure]

Figure 2: Violin plots with scattered particles as example for a fast switch in assigned model probability (cf. Fig. 11). The example refers to Findelgletscher (station FIN 1). Shown are (a) predictions of the individual models, (b) the ensemble prediction, and (c) the particle likelihood for two subsequent days. The weight of the individual models is given in Figure 11. Note that the ensemble is dominated by the OerlemansModel for the first day (left), and by the PelicciottiModel on the second day (right).

when not fixing $k$ but following a specific particle backwards in time, its model index does not change (see also our response to the comment on line 289). In this sense, we believe that our statement "There is no stochasticity in the evolution of $m_t$" is correct. To clarify this, we have rephrased l. 338 in the following way: "Because there is no stochasticity in the evolution of $m_t$ though, when the particle index $k$ is fixed,..."

*RC: Sec. 3.4 This section is essentially incomprehensible, with the section on proper scoring reading like it was pasted from a statistical methods paper. This being the Cryosphere, it's important to try to help your reader with some intuition as to what the CRPS actually means, and why its potential impropriety matters. A figure describing the metric might be useful, or perhaps a simple example de-scribing circumstances where the value is high or low. While the rest of the paper is still accessible not understanding CRPS, the analysis breaks down to 'big number bad, low number good,' which is unfortunate given that there is probably much more insight to be gained from the following sections.*
**AR:** We have simplified the paragraph and taken some statistics jargon out of it. We have also added the suggested example (but did not add the figure to not increase the manuscript's length further): "[...]It takes into account both the deviation of the median forecast from the actual observation (forecast reliability) and the spread of the forecast distribution (forecast resolution). This means that a forecast close to the observation median can still receive a poor Continuous Ranked Probability Score (CRPS) if the forecast distribution spread is high, and the other way around. Lower values of the CRPS correspond to better forecasts. The minimum value is zero, corresponding to a perfect, deterministic forecast of the observation.

The CRPS is defined as (Hersbach, 2000):

$$\text{CRPS} = \int_{-\infty}^{\infty} [P_f(b_{\text{sfc}}/\rho_{\text{bulk}} \cdot \rho_w) - H(b_{\text{sfc}}/\rho_{\text{bulk}} \cdot \rho_w - h(t,z))]^2 db_{\text{sfc}} \qquad (4)$$

where $P_f(\cdot)$ is the forecast mass balance cumulative probability distribution, and $H(\cdot)$ is the Heaviside function. The usual choice for $P_f$ is the weighted ensemble distribution of the predict particles, i.e. a discrete step function with jumps of height $w_{t-1,k}$ at the positions $\mathcal{H}(b_{\text{sfc}}(t,z)_k)$, where $b_{\text{sfc}}(t,z)_k$ are the prediction particles. Note that this setting does not account for the observation error of $h(t,z)$, implying that the score is not "proper", i.e. it does not always return the best value when the prediction distribution is the true distribution (Ferro, 2017, Brehmer and Gneiting, 2019). To obtain a proper score, one can use the forecast of the camera reading $h(t,z)$, which is the Gaussian mixture with weights $w_{t-1,k}$, mean values $\mathcal{H}(b_{\text{sfc}}(t,z)_k)$, and common variance $\sigma_\epsilon^2$. Despite being proper, it still has some theoretical shortcomings (Ferro, 2017). Since for our data the values of the two scores do not differ much, we use only the proper score in all results figures, but give also the value of the common CRPS in square brackets in the text."

*RC: Sec. 4.2.1 This section is quite unclear, specifically what the differences are that these include relative to the 'full' forecast.*

**AR:** We have avoided the wording "full" in this paragraph and changed the sentences to: 'We consider two types of reference forecasts: first, a forecast with (i) mean glacier-wide melt parameters as obtained from past calibration (Section 3.2) and (ii) the precipitation correction factor $prcp_{scale}$ constrained by the 2019 Glacier Monitoring Switzerland (GLAMOS) winter mass balance. Second, a forecast with a partially informed model including the same constraint for $prcp_{scale}$, but also a tuning of the melt parameters to reproduce one further intermediate point measurement. The latter measurement is the cumulative mass balance between September 2018 and 2019 at the mass balance stake closest to each camera (locations on Figure 1). Since there are up to four stake readings per glacier, we calculate single parameter sets tuned to reproduce all possible combinations of stake readings per glacier. This results in 19 CRPS values in total, for which we calculate the median. We also distinguish between the case in which the uncertainties in the meteorological inputs are taken into account and the case in which they are not.

Finally, we calculate the CRPS for the two reference forecasts by inserting two different values into the CRPS equation: (a) the mass balance of each day separately, and (b) the cumulative mass balance. Note that for the particle filter, there is no need to make this distinction. Indeed, the daily deviation from a mass balance observation also equals the deviation from the cumulative observation."

**RC:** *L435 Perhaps I missed it, but I can't find anything describing what the number in brackets means.*
**AR:** It is explained in l. 371, but indeed far from the first number occurring in this format. We have added another hint in the Results section (see also comment from Anonymous Referee #2 on this line): "(proper CRPS outside, non-proper CRPS inside the square brackets)".

**RC:** *Sec. 4.2.2 This section on cross-validation is very clear and good. Maybe it would be useful to comment on the temporal pattern evident in Figure 9, with CRPS in-creasing through time, but at different rates between different cross-validation folds.*
**AR:** This is indeed a very interesting feature. We have added the following sentences: "The temporal pattern evident in Figure 9 includes an increasing CRPS through time, but at different rates depending on the cross-validation subset. The individual pattern originates from (1) a stations' representativity for the given elevation band it is located in, (2) the combination of stations in the cross-validation subsets, and (3) cumulative error characteristics, since we observe cumulative mass balance over time. Station RHO 3, for example, can generally be modelled with lower errors compared to other stations. We speculate this being related to its location, which is in a relatively flat area with little crevasses. The other stations are instead either in the vicinity of crevasses (RHO 4) or influenced by shadows from the surrounding terrain, dark glacier surface or steep ice (RHO 1 and RHO 2). RHO 1 and RHO 2 also show that even neighboring stations can exhibit different melt. This affects the results of the cross-validation whenever one of these two stations is excluded from the training dataset."

**RC:** *L481 I don't understand where the '45 distinct model runs' come from. Also, what is a 'random coupling'?*
**AR:** We have clarified this in the text in the following way: "During the period preceding the installation of our cameras, we calculate mass balance with the parameters calibrated in Section 3.2. This results in about 45 distinct model runs, which we call "free model runs". We use this first period to provide initial conditions for the particle filter period, which lasts from the first camera setup on a respective glacier either until cameras are retrieved, or until the autumn field date is reached (whatever comes first). To achieve a connection between the free model run and the period during which the particle filter is used, we sample 10000 times from the initial conditions at the first camera setup date. We refer to this procedure as to "particle filtering without pre-selection (of initial conditions)". Not all free model runs have to be used, though: they can also be pre-selected based on the cumulative mass balance observed at the stakes closest to the camera stations. For this case, we select model runs that reproduce these observations within an estimated reading uncertainty of $\pm$ 0.05 m w.e. ("particle filtering with pre-selection"). The cumulative mass balances calculated with these two procedures are compared to the GLAMOS analyses [...]."

**RC:** *L495–496 I don't understand this sentence, nor why conditioning initial conditions on observations leads to poorer results.*
**AR:** This is something that we discussed extensively as well. The only reasons we can think of for why the conditioning can also lead to worse overall results are the following: 1) the mass balance stakes are several meters to hundreds of meters away from the camera installations, and are thus not "true" observations at the camera locations. This might result in the initial conditions being conditioned on biased observations with respect to the camera locations. 2) Either the CRAMPON and/or the GLAMOS uncertainties might be too small or too large. This can cause either of the analyses to be over-confident. 3) The

GLAMOS glacier-wide annual mass balances are interpolated from the stake readings using a model, and CRAMPON uses simplified geometries to calculate mass balances. If combined unluckily, this might lead to a stronger deviation from the GLAMOS annual mass balance when conditioned on point observations. Concretely, this means that because it was only possible to mount the cameras in our study on the lower 30% of the glacier surface area, they have a spatial bias and thus might not be representative for the vertical mass balance gradient. The three points are mentioned in the manuscript already, but have elaborated more on point (2) and (3): "Overall, the differences to the GLAMOS analyses can be explained by the (1) difference in the approaches used to calculate glacier-wide mass balances from point observations, (2) use of only 1-4 point observations located in the ablation zone and covering <30% of the glacier elevation range, compared to the complete network of 5-14 stakes over the entire elevation range used in the GLAMOS analyses, (3) lack of representativeness of the camera observations for the accumulation zone of the glaciers, i.e. biased vertical mass balance gradients, (4) lack of representation of individual winter accumulation measurements in our glacier model, or (5) a problem with representing the mass balance of the glacier with only one parameter set. Also note that 91-99% of the total uncertainty for the model runs with data assimilation stem from the period before the particle filter can be initialised, i.e. before the installation of the first camera station."

*RC: Figure 11 To emphasize earlier comments again, this pattern of mode collapse is strongly indicative of an over-confident likelihood operating in an M-open framework. It's well known that Bayesian inference only 'works' when the models are correctly specified. For Bayesian model averaging (which is what the particle filter is doing in a time dependent way), this still holds: because the true physics are not contained in the set of equations that the filter has available to pick from, yet this additional uncertainty is not explicitly specified, the filter hops between the model that fits the observations in the moment. While I don't expect any additional analysis, I think it would be appropriate to make this assumption explicit in the text, and to perhaps reference it when describing the fast switching between dominant models.*

**AR:** This is really a valuable inspiration for future work. We agree with the comment but would add that, in our view, it is not the likelihood being overconfident, but rather the prediction and thus the prior following from that. This is because we have chosen the observational error $\sigma_\epsilon$ conservatively. We could experiment with likely values for $\beta_t$, which would also depend on the meteorological input uncertainties, until the Particle Filter slowly converges towards one model. However, we believe that this is critical, since values for $\beta_t$ are volatile and tuning them would be subject to manual intervention. By implementing the procedure suggested for the lapse rates as our reply to the comment on l. 133, we have taken another source of uncertainty into account, which counteracts the "hopping" between models. In order to include an explicit error term $\beta_t$, we would have to specify a distribution also depending on the meteorological inputs. This would be a major undertaking. Regarding the request to clarify this in the text, we have made the following addition (together with the edits according to the comment on Eq. 16): L.307 "By introducing both the meteorological uncertainty $\eta$ and the parameter uncertainties, we shift the majority of the uncertainty contained in $\beta_t$ to these variables. Since the remaining uncertainty for $\beta_t$ is small and hard to quantify, we set $\beta_t = 0$ for simplicity. With this assumption, we neglect some additional uncertainty contained in $\beta_t$, being aware that this might lead to "jumps" in the temporal evolution of the model performance."

L. 515 "In our case, we believe that the ensemble prior might be overconfident on average, since we have chosen the observational error conservatively, i.e. we have chosen the largest errors emerging in the round robin experiment (Section 2.1.2). This would lead to a model preferably obtaining high weights, which has already dominated on the previous days. However, when the likelihood is overconfident or there is strong evidence that a previously well performing model now performs worse, the filter might switch back and forth between individual models that best describe the observations. We accept this model dominance and the fast switching as a sign that the overall ensemble performance is improved."

*RC: Figure ?? Two things that are missing from the paper are time series' of state and parameter distributions. It would be very interesting to see the evolution of un-certainty in the predictions away from observations, and also to see how quickly parameters change or revert to the mean.*

**AR:** We agree that this is very interesting. We have added a figure and a small paragraph discussing the aspect of parameter evolution in the new manuscript: "'Figure 13 shows the evolution of the distribution of individual model parameters during the assimilation period. The example refers to Findelgletscher. Three phases of quick parameter changes can be observed: First, the parameters change rapidly on the first days of the assimilation period. This means that the prior parameter distributions do not match the exact parameter distributions needed to model the mass balance at the camera locations. This is due to both the calibration time span (seasonal calibration vs. daily application) and the low sample size of the calibrated parameters. A second rapid change can be observed after the second camera has been switched

[Figure]

Figure 3: Temporal evolution of the various model parameters for Findelgletscher. Shown are the sample means (lines) and the standard deviation (bands). Note that for the OerlemansModel, parameter $c_0$ is adjusted to fit on the same scale as $c_1$.

on, i.e. on July 24th, 2019. Here, an adjustment in the parameters is needed in order to accommodate the mass balance at both stations equally well. The third rapid change starts when ablation at station FIN 1 is highest, but when radiation and temperature are not at their maximum. Here, the change might be due to the model being forced to yield high ablation rates despite only moderate meteorological forcing. This shows the advantage of employing the model ensemble as opposed to e.g. a single model with deterministic parameters: the ensemble also reproduces system states which cannot be explained by the uncertain meteorological input."

Regarding the evolution of the mass balance state, we have added Figure 4. However, to not extend the main text any further, we have put this figure in the Appendix and give a reference in the main text.

[Figure]

Figure 4: Temporal evolution of the ensemble mass balance state at stations FIN-1 and FIN-2. In the top two panels, the evolution of the mean and standard deviation of the filter (black lines and yellow shaded area) around the centered observations (blue lines and blue shaded area) is shown. In the bottom panel the mean deviation of the filter from the observations at both stations is shown.

**New references**

- Bernardo, J. M., & Smith, A. F. (2009). Bayesian theory (Vol. 405). John Wiley & Sons.

- Zellner, A. (1986). On assessing prior distributions and Bayesian regression analysis with g-prior distributions. Bayesian inference and decision techniques.

**Author response to the review of Anonymous Reviewer #2**

Johannes Landmann and co-authors

June 2021

Dear Anonymous Referee #2,
Dear Editor,
We would like to thank Anonymous Referee #2 for the valuable review of our manuscript. We appreciate the constructive comments and the positive feedback regarding the overall context of the study. We address the comments point-by-point answers to the individual remarks and questions.
We have also taken the opportunity to further streamline out wording, by simplifying a number of sentences and paying attention to the overall text coherence. We are convinced that this has resulted in a much more accessible manuscript as compared to the first submission.
Best regards,
Johannes Landmann and co-authors

**– Major comments and questions –**

**RC:** *What made you choose a particle filter as DA method as opposed to more traditional methods (e.g. variational methods and ensemble Kalman filter). It would complement the work if you discussed why a PF suits this problem.*
**AR:** The Particle Filter (PF) is a generic data assimilation method that can handle all kinds of model distributions, also non-linear ones. We have chosen the PF since we know that the distributions we deal with are not always Gaussian. To not extend the already long introduction any further, we have added these explanations at l. 250, where we introduce the Particle Filter: "Especially when temperatures are around the melting point, the system becomes non-linear, since melt occurs above but not below this point. As a consequence, the distributions we deal with are not necessarily Gaussian. The facts that (a) the temperature chosen to parametrize the melting point is not the same for all four models, (b) the individual model prior distributions are combined to obtain the ensemble prediction, and (c) there can also be accumulation contributing to the overall mass balance, add further complexity. We do not use other data assimilation approaches, such as variational methods or Ensemble Kalman filtering, because variational methods encounter difficulties when dealing with non-Gaussian priors (van Leeuwen et al., 2019), whilst the Ensemble Kalman Filter in its original form is not designed for multi-model applications as we use in our case. Overall, particle filtering is a very flexible, generalizable, and readily implementable data assimilation method."

**RC:** *Line 105. If I understood well, the observations are of a cumulative quantity. In this case, do observation errors need to consider time auto-correlations?*
**AR:** This is a justified question. The observations are of a cumulative quantity, but not auto-correlated. This is because mass balance at a given time is not inferred by summing the individual, sub-daily readings up to that time. Rather, the mass balance of a given point in time is given by one single reading, i.e. by the reading of the total height change at the stake up to that time. This is made possible by the colour coding used on the stake: the colour sequence is chosen such that any location of the stake can uniquely be identified as soon as four coloured stripes are visible. To clarify this, we have added the following text at L.107: "We expect the observation errors to be uncorrelated in time, since every reading is independent from the previous one."

**RC:** *Line 255. I was a bit confused on where the uncertainties of the input variables are represented. Are they represented in the model error beta, in the observation error epsilon (as mentioned in line 258), or both?*
**AR:** The observational error from the camera readings is represented in the observation error $\epsilon$, as stated in l. 257. The model input errors are considered to be contained in the model error $\beta_t$, as stated in l. 258.

We have clarified this by replacing "[...] ($\beta_t$) can also represent uncertainties in model input variables" with "[...] ($\beta_t$) should include uncertainties stemming from the model input variables" in l. 258.

**RC:** *The use of the PF in a multi-model ensemble context is quite interesting, especially since each model has different parameters one is trying to estimate. Is there previous work in this regard? Could you provide some references?*
**AR:** Currently, we are not aware of other studies that have applied particle filtering in a multi-model ensemble related to glacier mass balance. However, we have added some references to applications in other contexts:

- Kreucher, C., Hero, A., & Kastella, K. (2004, March). Multiple model particle filtering for multitarget tracking. In Proceedings of the Twelfth Annual Workshop on Adaptive Sensor Array Processing.

- Ristic, B., Arulampalam, S., & Gordon, N. (2004). Beyond the Kalman filter: Particle filters for tracking applications (Vol. 685). Boston: Artech house.

- A. Saucan, T. Chonavel, C. Sintes and J. Le Caillec, "Interacting multiple model particle filters for side scan bathymetry," 2013 MTS/IEEE OCEANS - Bergen, 2013, pp. 1-5, doi: 10.1109/OCEANS-Bergen.2013.6608125.

- Wang, R., Work, D. B., & Sowers, R. (2016). Multiple model particle filter for traffic estimation and incident detection. IEEE Transactions on Intelligent Transportation Systems, 17(12), 3461-3470.

The revised text reads: "We are not aware of mass balance studies that have applied a multi-model ensemble based on a particle filter with the resampling methods we propose, although multi-model particle filters have been used for other applications (e.g. Kreucher et al., 2004, Ristic et al., 2004, Saucan et al., 2013, Wang et al., 2016)."

**RC:** *When discussing the particle filter, you introduce the concept of 'minimum contribution' for some particles. This is taken into account when weighting, as explained in appendix 2. There is a comment saying that the original weights are preserved 'in average'. Could you elaborate more on this statement?*
**AR:** Only preserving the weights on average is common for resampling procedures: when a particle performs poorly, it obtains a weight of zero and disappears. If a particle $x_{t,k}$ has a weight $w_{t,k}$, then it is chosen $N \cdot w_{t,k}$ times in the resampling. This particle then has the weight 1/N times the "number how often it has been resampled", so on average $w_{t,k}$. The same is true when resampling within a model: a particle $x_{t,k}$ with model index $j$ is resampled on average $N_{t,j} \cdot w_{t,k}/\pi_{t,j}$ times. After resampling it has the weight $\tilde{w}_{t,k}$ times "number how often it was resampled", so on average $w_{t,k}$. This statement is made in Equations B2 and B3, which are found in Appendix B.

**RC:** *Equation 21. How are $\mu_0$ and $\Sigma_0$ chosen?*
**AR:** We choose $\mu_0$ and $\Sigma_0$ from the parameter statistics obtained from the calibration procedure in section 3.2. We have added a phrase stating where $\mu_0$ and $\Sigma_0$ originate from. Revised text: "$\vec{\mu}_0$ and $\vec{\Sigma}_0$ are the prior mean and the prior covariance of $\vec{\theta}$ at the starting time $t_0$, which we determine from the calibration procedure described in section 3.2,[...] "

**– Minor comments and questions –**

**RC:** *The title mentions 'mass balance observations', whereas the observations are of surface elevation.*
**AR:** Our intention was to simplify the wording for the reader. As explained in our response to the comment on the manuscript title by Reviewer #1, we have changed the title as follows: "Assimilating near real-time mass balance stake readings into a model ensemble using a particle filter"

**RC:** *Line 96. It is mentioned that the camera images are read 'manually' to obtain the daily cumulative surface height change. Is it literally reading the marks from the ablation pole? How could this be automated to be applied to more places?*
**AR:** Yes, we have read the marks manually. We are working on a procedure to automate this though, so that operational runs that we plan for the future won't require manual interventions. In this respect, see our EGU2021 abstract (https://doi.org/10.5194/egusphere-egu21-7663) and our GitHub repository (https://github.com/leosold/TOAST).

**RC:** *Line 127. Can you say more about the 0.2 degree resolution? How does this compare with other products? Is it high or low resolution?*

**AR:** At Swiss latitudes, 0.2 degrees corresponds to a resolution of about 2km. Compared to Global Climate Models or Regional Climate Models, which are sometimes used to force glaciological models directly, this is a very high resolution. We have specified this in the revised text: "... which for Switzerland corresponds to a horizontal resolution of about 2 km."

**RC:** *Figure 7. I think making the vertical axis larger for panels a and b could make the figure easier to read.*

**AR:** As a response to the request, we have doubled the extent of these two axes:

[Figure]

**RC:** *Figure 8. The individual circles are difficult to see. Please make the circumferences thicker, and maybe increase the size of the figure.*

**AR:** We have increased the line thickness of the circles to improve the visibility and we have also increased the figure size:

[Figure]

**RC:** *In pages 22-24 (approximately) there are several places where a quantity is written followed by () or []. It was not clear to me what the quantities in the parenthesis are, and why there are two styles.*

**AR:** It was explained in l. 371 that we use the square brackets to add the non-proper Continuous Ranked Probability Score (CRPS) into the text, but apparently it needs to be repeated in the Results section. Moreover, the round brackets we use in our notation are to be understood as annotations in the way they are commonly used. We have added a reminder at the first occurrence of the square brackets: "(proper CRPS outside, non-proper CRPS inside the square brackets)".

**- Typos and corrections -**

*RC: Line 109. ... because it can happen that the camera construction sinks...→ ... because the camera construction can sink. (easier to read).*
**AR:** We have changed this as suggested.

*RC: Line 118. melt during night → nighttime melting*
**AR:** We have changed this as suggested.

*RC: Line 190 on. When mentioning the models in an itemised list, start the sentences with capital letter.*
**AR:** We have changed this as suggested.

*RC: Figure 5. Some of the words in the labels are split into two lines*
**AR:** This was a tradeoff between font size and visual appearance. We have reformatted the figure so that the individual words are better readable:

[Figure]

*RC: In the title of table 2 it should say 'standard deviations' instead of 'covariances'.*
**AR:** We thank the reviewer for noticing this. We have corrected this.